# Chemical reprogramming ameliorates cellular hallmarks of aging and extends lifespan

Lucas Schoenfeldt [1,2,3], Patrick T Paine[1,3], Sara Picó[1,3], Nibrasul H Kamaludeen M[1], Grace B Phelps[1,2], Calida Mrabti[1], Gabriela Desdín-Micó [1], María del Carmen Maza[1], Kevin Perez [1,2 ✉] & Alejandro Ocampo [1,2 ✉]

## Abstract

The dedifferentiation of somatic cells into a pluripotent state by cellular reprogramming coincides with a reversal of age-associated molecular hallmarks. Although transcription factor induced cellular reprogramming has been shown to ameliorate these aging phenotypes in human cells and extend health and lifespan in mice, translational applications of this approach are still limited. More recently, chemical reprogramming via small molecule cocktails have demonstrated a similar ability to induce pluripotency in vitro, however, its potential impact on aging is unknown. Here, we demonstrated that chemical-induced partial reprogramming can improve key drivers of aging including genomic instability and epigenetic alterations in aged human cells. Moreover, we identified an optimized combination of two reprogramming molecules sufficient to induce the amelioration of additional aging phenotypes including cellular senescence and oxidative stress. Importantly, in vivo application of this two-chemical combination significantly extended *C. elegans* lifespan and healthspan. Together, these data demonstrate that improvement of key drivers of aging and lifespan extension is possible via chemical-induced partial reprogramming, opening a path towards future translational applications.

**Keywords** Aging; Cellular Reprogramming; Chemical Reprogramming; Epigenetics; Lifespan
**Subject Categories** Autophagy & Cell Death; Metabolism; Pharmacology & Drug Discovery

## Introduction

Biological aging is a global process associated with a loss of homeostasis and functional decline across cellular and physiological systems, leading to the development of age-associated chronic diseases and finally death (Rando and Chang, 2012; Kennedy et al, 2014). For this reason, a significant increase in human life expectancy over the last decades has resulted in an extended period of life spent in morbidity (Garmany et al, 2021). As aging is a key risk factor for chronic diseases such as cardiovascular disease or neurodegenerative disorders, new therapeutic strategies that target aging are now under intense investigation (Niccoli and Partridge, 2012; Mahmoudi et al, 2019). Several hallmarks of aging, including epigenetic dysregulation, genomic instability, cellular senescence, and stem cell exhaustion, have been identified as potential targets for such an optimized therapeutic strategy (López-Otín et al, 2013). Current interventions that target aging include cellular reprogramming, dietary restriction and related mimetics, systemic blood factors, and senolytics. Among these, cellular reprogramming offers a unique prospective for its ability to reset the epigenome and restore multiple aging hallmarks (Conboy et al, 2005; Baker et al, 2011; Lapasset et al, 2011; Longo et al, 2015; Ocampo et al, 2016b; Olova et al, 2019).

During development, cellular reprogramming induces zygotic and primordial germ cell formation following a dramatic chromatin reorganization to create totipotent and pluripotent cells free of aged molecular defects, demonstrating that both cell identity and age are reversible (Seisenberger et al, 2013; Kerepesi et al, 2021). Importantly, this manipulation of cell identity has been recapitulated in vitro by several methods, including somatic cell nuclear transfer, forced expression of transcription factors, and most recently treatment with small molecules (Gurdon, 1962; Takahashi and Yamanaka, 2006; Hou et al, 2013). More precisely, pluripotent stem cells can be generated from mouse somatic cells following 40 days of sequential seven small-molecule treatments (Hou et al, 2013).

Although restoration of aged phenotypes such as telomere length, mitochondrial function, proliferation, and transcriptomic signature in vitro was demonstrated over a decade ago, application of cellular reprogramming in vivo was initially proven unsafe due to the loss of cellular identity leading to tumor and teratoma formation (Lapasset et al, 2011; Abad et al, 2013). To overcome this issue, in vivo partial reprogramming by short-term cyclic induction of Oct4, Sox2, Klf4, and c-Myc (OSKM) was a critical advance as it avoided the detrimental loss of cellular identity. Importantly, this limited cyclic expression of OSKM was sufficient to ameliorate multiple aging hallmarks and extend the lifespan of a progeroid mouse strain (Ocampo et al, 2016b). Improved regenerative capacity and function has also been demonstrated following

[1]Department of Biomedical Sciences, Faculty of Biology and Medicine, University of Lausanne, Lausanne, Vaud, Switzerland. [2]EPITERNA SA, Epalinges, Switzerland. [3]These authors contributed equally: Lucas Schoenfeldt, Patrick T Paine, Sara Picó. ✉E-mail: kevin@epiterna.com; alejandro.ocampo@unil.ch

therapeutic application of cellular reprogramming in multiple tissues and organs including the intervertebral disc, heart, skin, skeletal muscle, liver, optic nerve, and dentate gyrus (Ocampo et al, 2016b; Kurita et al, 2018; de Lázaro et al, 2019; Lu et al, 2020; Rodríguez-Matellán et al, 2020; Chen et al, 2021; Cheng et al, 2022; Hishida et al, 2022). Furthermore, several groups have demonstrated the ability to restore multiple aging phenotypes and reset the epigenetic clock utilizing translational non-integrative methods such as modified mRNAs encoding for six transcription factors (OSKM + Lin28 and Nanog) or adeno-associated virus for expression of three factors (OSK)(Sarkar et al, 2020; Lu et al, 2020). While promising, methods that require transcription factor expression face significant barriers for their clinical translation, such as risk of tumorigenicity and low delivery efficiency (Abad et al, 2013; Ohnishi et al, 2014). In this line, c-Myc and Klf4 have been identified as proto-oncogenes, while Oct4 and Sox2 are highly expressed in a variety of human cancers (Klimczak, 2015). For this reason, the clinical application of in vivo reprogramming may require further development.

Most recently, small-molecule cocktails have been shown to produce chemical-induced pluripotent stem cells (ciPSCs) from mouse and human somatic cells (Hou et al, 2013; Guan et al, 2022a). These reprogramming compounds fall broadly into three categories, including epigenetic, cell signaling, and metabolic modulators (Knyazer et al, 2021). Importantly, small-molecule reprogramming and OSKM expression share the ability to overcome multiple reprogramming barriers while retaining a distinct cell fate trajectory (Zhao et al, 2015; Haridhasapavalan et al, 2020). To date, the effects of chemical reprogramming on aging hallmarks and lifespan have not been investigated. Considering both the rejuvenating effects of partial reprogramming by short-term expression of OSKM and the ability of small-molecule cocktails to induce pluripotency, we proposed the use of chemical-induced partial reprogramming for the amelioration of aging phenotypes.

Here, we report that short-term treatment of human cells with seven small molecules (7c), previously identified for their capacity to induce pluripotent stem cells, leads to the improvement of molecular hallmarks of aging. In addition, we show that an optimized cocktail, containing only two of these small molecules (2c), is sufficient to restore multiple aging phenotypes, including genomic instability, epigenetic dysregulation, cellular senescence, and elevated reactive oxygen species. Finally, in vivo application of this 2c reprogramming cocktail extends both lifespan and healthspan in C. elegans.

# Results

## Chemical-induced partial reprogramming significantly improves aging hallmarks in aged human fibroblasts

Multiple hallmarks of aging can be ameliorated following partial cellular reprogramming by expression of the Yamanaka factors (OSKM) in vitro and in vivo (Ocampo et al, 2016a). On the other hand, although chemical reprogramming with seven small molecules (7c) has been shown to generate chemically induced pluripotent stem cells (Hou et al, 2013), whether chemical-induced partial reprogramming is also able to restore aged phenotypes is unknown. Therefore, we sought to determine the effect of short-

term 7c treatment on aging phenotypes in primary aged human fibroblasts (Fig. 1A). Differently from the initial full reprogramming protocol (Hou et al, 2013), we decided to perform partial reprogramming treatment using the seven chemicals continuously for a duration of 6 days. Specifically, we asked whether chemical-induced partial reprogramming could improve multiple hallmarks of aging, including DNA damage, heterochromatin loss, cellular senescence, and reactive oxygen species (ROS) in vitro. Towards this goal, primary human fibroblasts isolated from aged dermal tissue samples were treated for 6 days with a 7c cocktail including: CHIR99021, DZNep, Forskolin, TTNPB, Valproic acid (VPA), Repsox, and Tranylcypromine (TCP). Notably, the levels of the DNA damage marker γH2AX were significantly decreased in aged cells after treatment (Fig. 1B). Interestingly, a decrease in γH2AX levels was also observed when 7c was added for 6 days to aged cells that were pretreated with the DNA damage inducing agent doxorubicin for 2 days, indicating a significantly improved DNA damage response upon 7c treatment (Fig. 1C). Thus, we determined that short-term 7c treatment improves DNA damage in primary aged human fibroblasts.

Epigenetic dysregulation and heterochromatin loss are key molecular markers of aging (Haithcock et al, 2005; Scaffidi and Misteli, 2006; Ni et al, 2012; Brunet and Berger, 2014; Djeghloul et al, 2016; Kane and Sinclair, 2019). For this reason, we next examined the effect of 7c treatment on the constitutive and facultative heterochromatin marks H3K9me3 and H3K27me3. Our results show that 6 days of 7c treatment significantly increased the constitutive heterochromatin mark H3K9me3 in aged human fibroblasts (Fig. 1D). In aged cells, previous work has shown that the facultative heterochromatin mark H3K27me3 is decreased at the senescence-associated p16 gene locus (CDKN2A) leading to increased expression, cell cycle arrest, and senescence (Dhawan et al, 2009). Interestingly, we observed that H3K27me3 was significantly increased after 6 days of 7c treatment (Fig. 1E). Next, as cellular senescence has been shown to be a key driver of aged tissue dysfunction and ablation of senescent cells has been demonstrated to extend health and lifespan in mice (Baker et al, 2011), we determined the impact of chemical-induced partial reprogramming on senescence-associated gene expression. Interestingly, senescence-associated and age-related stress response genes including p21, p53, and Gadd45b were downregulated upon 7c treatment following 6 days of treatment (Fig. 1F). In addition, we serially passaged aged human fibroblasts for 28 days to promote replicative-induced senescence (RIS) in the presence of continuous 7c treatment. Importantly, the 7c-treated group showed a downregulation of the senescence-associated cell cycle genes p21 and p53 relative to untreated controls (Appendix Fig. S1A). On the other hand, we noted that the senescence-associated secretory cytokine IL6 was significantly upregulated upon long-term treatment with 7c and therefore could not conclude a positive effect of 7c treatment on senescence (Fig. 1F and Appendix Fig. S1A).

Next, further characterization of the transcriptomic effects of 7c treatment for 6 days was performed by bulk RNA sequencing on aged fibroblasts treated with either 7c or vehicle control. Principal component analysis (PCA) showed that 7c-treated cells clustered separately from the control group, indicating that a distinct transcriptomic profile emerges following chemical-induced partial reprogramming (Fig. 1G). Gene ontology (GO) enrichment analysis revealed that developmental processes were significantly

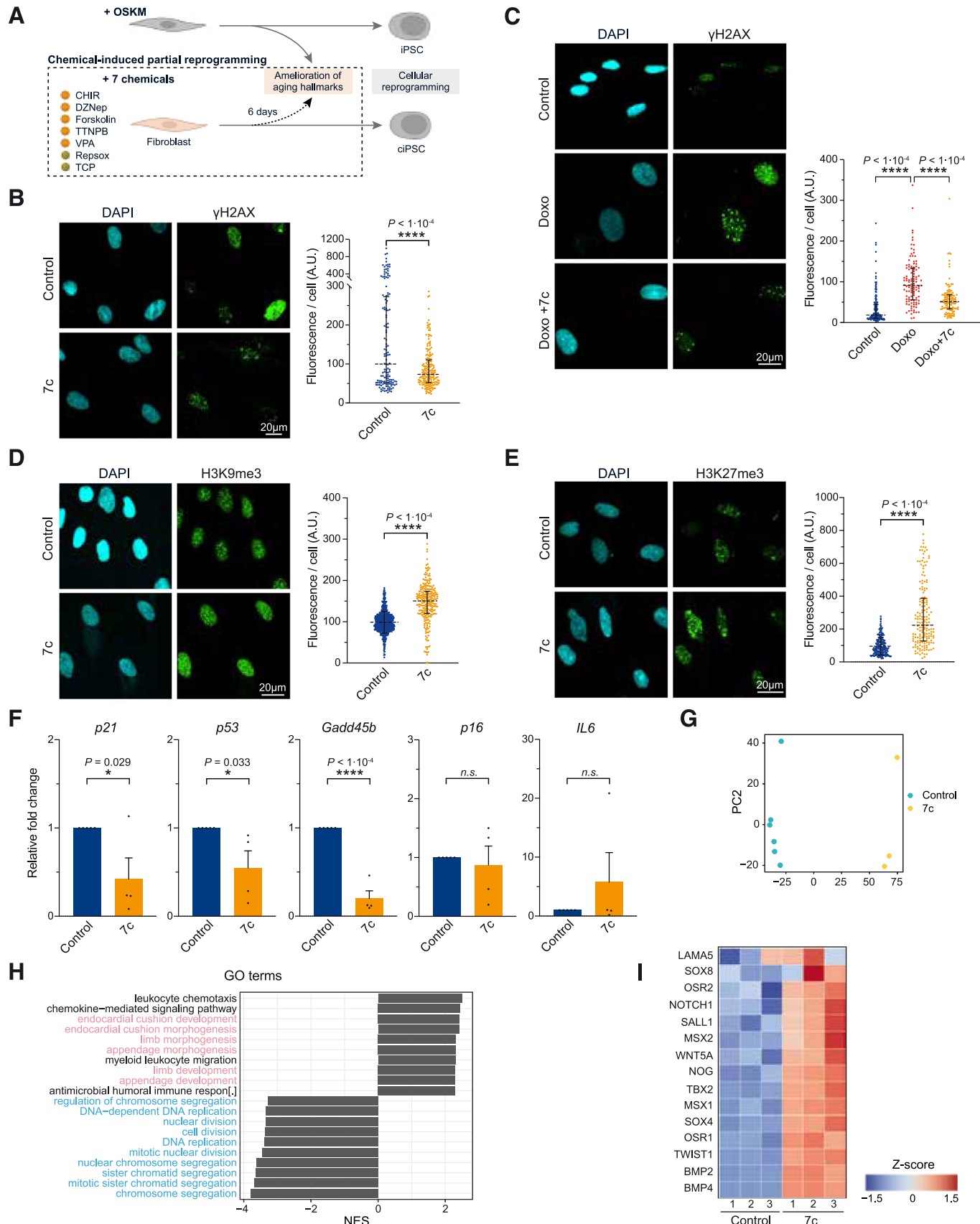

**Figure 1. Chemical-induced partial reprogramming significantly improves aged hallmarks in aged human fibroblasts.**

(A) Schematic representation of chemical-induced partial reprogramming via 6 days of treatment with the seven chemicals previously shown to induce mouse chemical iPSC (Hou et al, 2013). (B, C) Immunofluorescence and quantification of γH2AX following 7c treatment (B) or Doxorubicin (100 nM) and 7c treatment (C). (D, E) Immunofluorescence and quantification of H3K9me3 (D) and H3K27me3 (E) following 7c treatment. (F) mRNA levels of senescence-associated and age-related stress response genes in the *p53* tumor suppressor pathway following 7c treatment (6 days). (G) Principal component analysis (PCA) of control (blue) and 7c-treated (orange) fibroblasts. (H) Gene ontology (GO) enrichment analysis following 7c treatment with developmental (in pink) and cell cycle (in blue) pathways highlighted. (I) Heatmap showing the expression pattern of differentially expressed genes associated with developmental pathways following 7c treatment. Data were median ± IQR (B–E), mean ± SEM (F). (B, C) $n = 2$, (D–I) $n \geq 3$. Statistical significance was assessed by comparison to untreated control using a paired two-tailed *t*-test (B–F). NES normalized enrichment scores. Source data are available online for this figure.

upregulated following 7c treatment relative to control, whereas mitosis and cell proliferation programs were significantly downregulated (Fig. 1H). Interestingly, numerous cellular reprogramming, stem cell, and self-renewal genes within the GO term developmental pathways were significantly upregulated following 7c treatment compared to control including *WNT5A*, *NOTCH1A*, *SOX4*, *SALL1*, *NOG*, and *BMP4* indicating that 7c induced a shift towards a developmental associated transcriptomic profile (Fig. 1I; Appendix Fig. S1B).

Since our RNA-seq analysis indicated a downregulation in mitosis and proliferation programs, we next evaluated the effect of 7c treatment on cellular proliferation. In agreement with these results, we observed that 7c significantly decreased cell density based on MTS assay (Fig. 2A). This observation was confirmed by a strong decrease in the proliferation associated marker Ki67 in cells treated with 7c (Fig. 2B). Subsequently, to determine whether the effect of 7c on proliferation was dose-dependent, a serial dilution assay of 7c treatment was performed. Notably, different concentrations of 7c continued to impair proliferation (Appendix Fig. S2A). Overall, these results validate our transcriptomic findings, indicating that mitosis and proliferation-related programs are downregulated upon 7c treatment.

Next, in order to investigate whether the decrease in DNA damage upon 7c treatment was independent of cell cycle impairment, we tested the effect of 7c treatment under nonproliferative conditions in the presence of low-serum culture conditions (i.e., 1% FBS culture media). Remarkably, regardless of growth conditions and proliferation, 7c treatment still induced a reduction in γH2AX levels (Fig. 2C), suggesting that the impact of 7c on DNA damage is independent of its effects on proliferation. In addition, to gain further insight into the metabolic changes induced upon 7c treatment, we investigated the levels of reactive oxygen species (ROS), which are associated with mitochondrial function, cellular stress and DNA damage (Shields et al, 2021). Notably, a significant increase in reactive oxygen species (ROS) in aged fibroblasts was observed upon 7c treatment (Fig. 2D). Thus, although 7c treatment can decrease γH2AX levels, we find that it also leads to impaired proliferative capacity, even at low concentrations, and an upregulation of ROS. Taken together, chemical-induced partial reprogramming via 7c treatment in aged human fibroblasts lacks the multiparameter rejuvenation associated with OSKM-induced reprogramming. In particular, 7c treatment results in the improvement of several key aging phenotypes such as DNA damage, epigenetic dysregulation, and senescence markers, while at the same time leading to an impairment of proliferation, increased ROS, and upregulation of *IL6*. These results suggest that further optimization of chemical reprogramming may be required.

## A reduced reprogramming cocktail efficiently improves multiple molecular hallmarks of aging

In order to create an optimized cocktail for the amelioration of age-associated phenotypes by chemical reprogramming, we first sought to remove compounds with deleterious effects, while still retaining the three key functional categories of chemical reprogramming compounds: epigenetic modifiers, cell signaling modifiers, and metabolic switchers (Knyazer et al, 2021). In accordance with Knyazer *et al*, Repsox is established as a small-molecule inhibitor targeting the TGFβR-1/ALK5 signaling pathway, aligning with its functional classification. Moreover, Guo et al, demonstrated through RNA sequencing analysis a significant upregulation of DEGs associated with metabolic processes in Repsox-treated fibroblasts, reinforcing its relevance in chemical reprogramming, as also previously suggested by our group (Paine et al, 2024). On the other hand, TCP is an epigenetic modifier, which inhibits histone demethylase LSD1 activity, as outlined by Knyazer et al.

First, using an MTS assay, we observed that while cell survival and proliferation were unaffected or enhanced by Repsox or TCP treatment respectively, proliferation was impaired by CHIR99021, DZNep, Forskolin, TTNPB, and VPA at high concentrations, in agreement with previous publications and suggesting their removal (Appendix Fig. S2B) (Paine et al, 2024; Wu et al, 2006; Jung et al, 2008; Rodriguez et al, 2013; Girard et al, 2014). In addition, DZNep has a known *S*-adenosylhomocysteine (SAH) hydrolase-mediated inhibitory effect on the H3K27 methyltransferase EZH2, further supporting its exclusion (Girard et al, 2014). The remaining compounds, TCP and Repsox, met the selection criteria for chemical reprogramming functional categories and were thus selected. Therefore, we next treated aged human fibroblasts with this reduced two-chemical cocktail (2c) for 6 days to determine its effect on aging hallmarks.

Strikingly, similar to our previous results with 7c, γH2AX levels were significantly decreased upon 2c treatment (Fig. 3A). Furthermore, improvement on yH2AX levels was observed when cells were treated with 2c following addition of the DNA damaging agent doxorubicin (Appendix Fig. S3A). Moreover, 2c significantly increased both H3K9me3 and H3K27me3 levels (Fig. 3B,C). Taken together, these data indicate that 2c treatment improves DNA damage and heterochromatin marks similar to 7c treatment.

Next, we sought to determine the impact of 2c treatment for 6 days on cellular senescence in both a genotoxic stress-induced senescence model using doxorubicin application, and RIS after multiple passages. First, doxorubicin induced senescent cells pretreated with 2c showed a significant decrease in senescence-associated beta-galactosidase (SA-beta-gal) levels and *p21* gene expression compared to untreated controls (Fig. 3D,E).

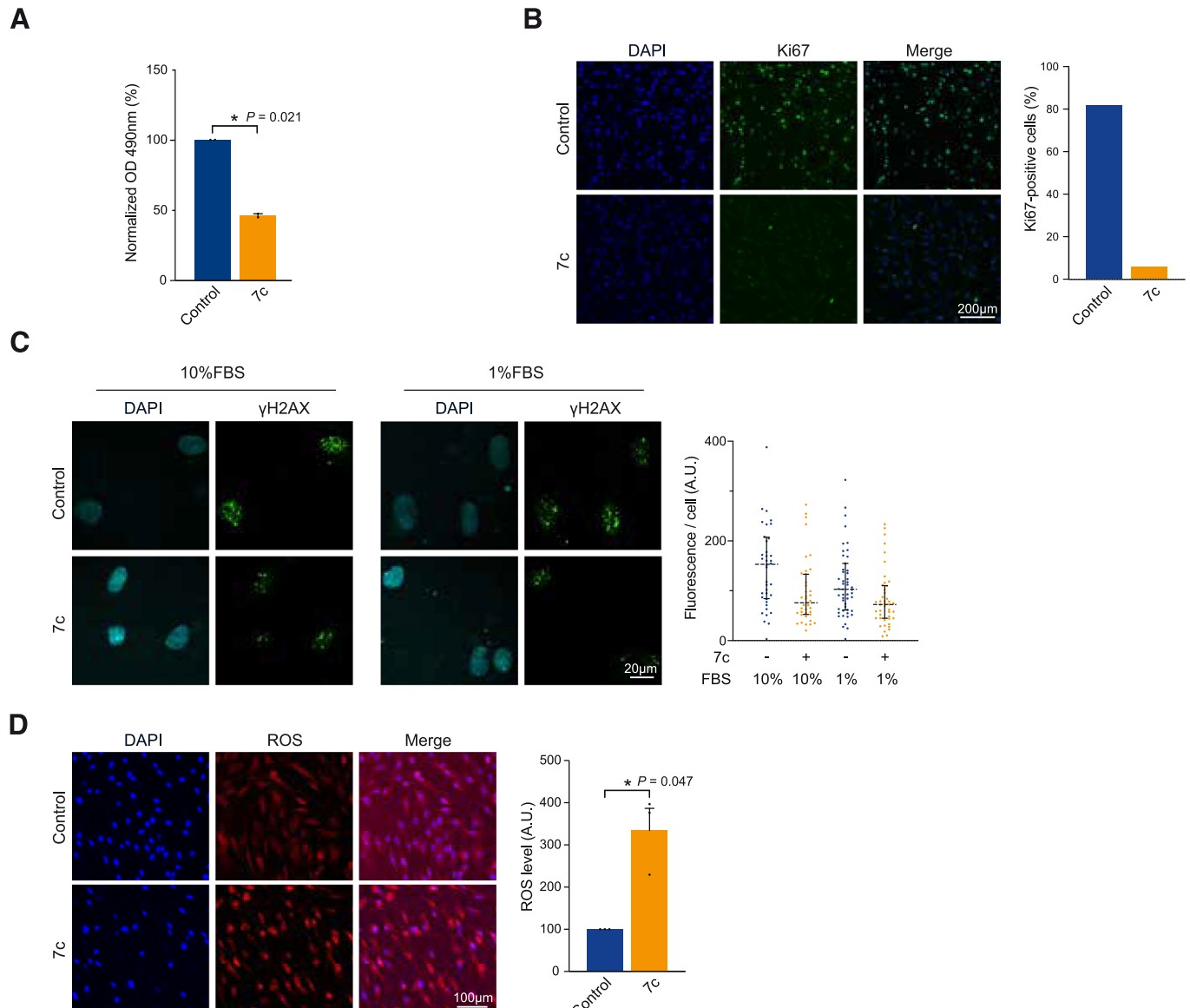

**Figure 2. Chemical-induced partial reprogramming via 7 chemicals does not fully induce multiparameter rejuvenation.**

(A) MTS quantification of cell density following 7c treatment until confluence. (B) Immunofluorescence and quantification of Ki67 following 7c treatment (6 days, "6D"). (C) Immunofluorescence and quantification of γH2AX following 7c treatment (6D) in proliferative (10% FBS) and non-proliferative (1% FBS) conditions. (D) Fluorescence detection and quantification of reactive oxygen species (ROS) following 7c treatment (6D). Data were mean ± SEM (A, D), median ± IQR (C). (A, B) $n = 2$, (C) $n = 1$, (D) $n \geq 3$. Statistical significance was assessed by comparison to untreated control using a paired two-tailed $t$-test (A, D). OD optical density. Source data are available online for this figure.

Interestingly, 2c significantly decreased SA-beta-gal levels only when added prior to induction, indicating a protective rather than senolytic effect upon genotoxic treatment (Appendix Fig. S3B). Interestingly, a similar observation was described in mice, where a short 6-day induction of OSKM prior to myocardial infarction (MI) significantly restored cardiac function, while post-MI treatment showed minimal improvement (Chen et al, 2021). Moreover, in our RIS model of long-term treatment, SA-beta-gal levels were significantly decreased in aged fibroblasts with continuous 2c treatment (Fig. 3F). In addition, senescence-associated and age-related stress response genes p21, p53, and IL6, were also downregulated upon 2c treatment after 6 days or 29 days of treatment (Fig. 3G). Taken together, these data show that 2c treatment reduces cellular senescence and significantly decreases IL6 levels in contrast to 7c treatment.

Next, as the 7c treatment previously led to impaired proliferation and increased ROS levels, we sought to determine the impact of 2c on these cellular phenotypes. Importantly, 2c treatment had only a mild effect on cell proliferation compared to the 50% decrease induced by 7c treatment (Fig. 3H). In addition, in clear contrast to the impact of 7c, 2c treatment significantly decreased ROS levels in aged fibroblasts, indicating that 2c can markedly

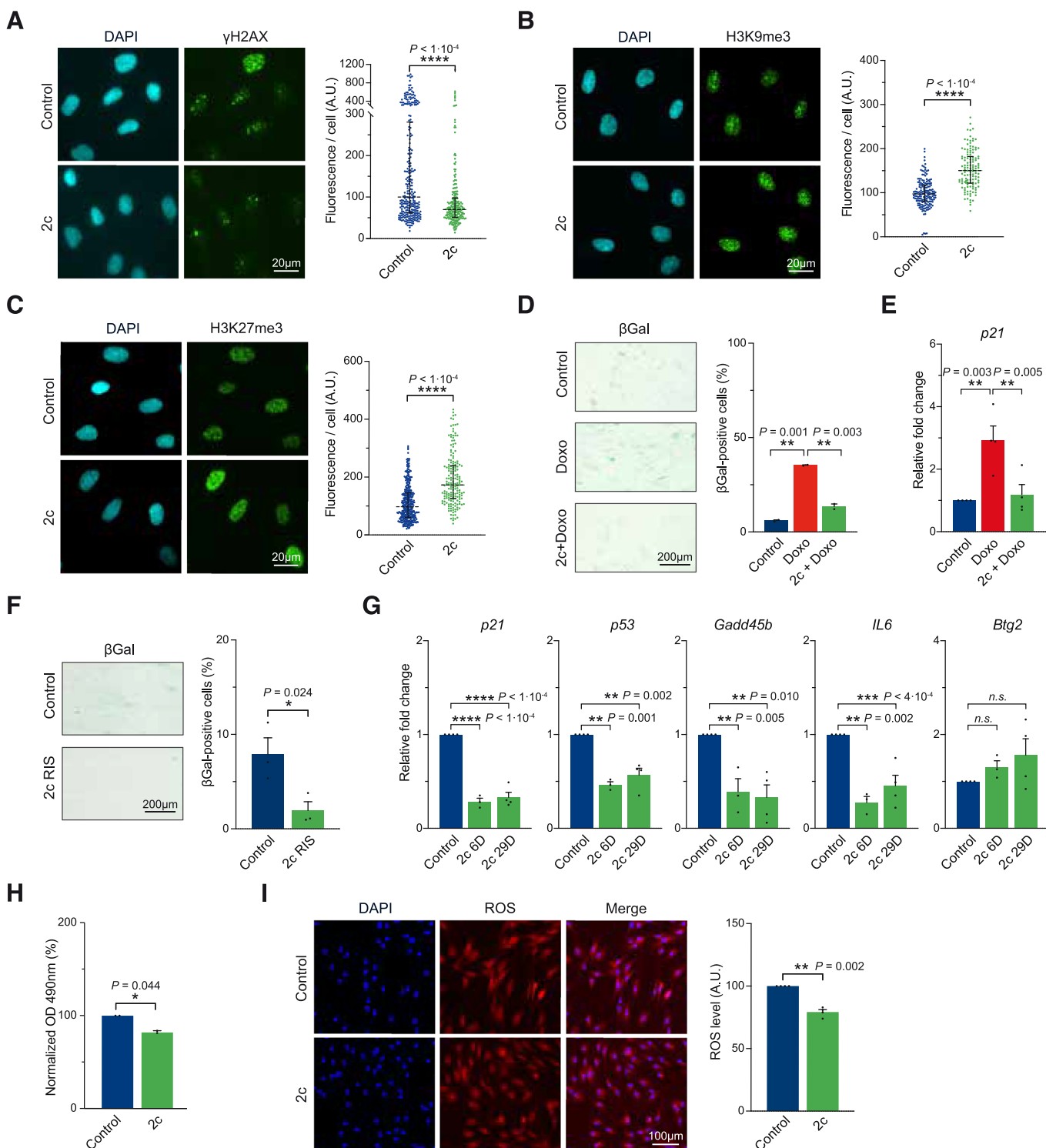

improve cellular homeostasis (Fig. 3I). Similarly, we observed an improvement in ROS levels with 2c treatment when cells were co-treated with the pro-oxidant Antimycin A (Appendix Fig. S3C). Taken together, these data indicate that 2c is an optimized cocktail capable of improving multiple age-related markers in vitro.

Additionally, as cellular reprogramming has been previously described in keratinocytes using OSKM, we sought to determine

whether the amelioration of age-associated phenotypes with our reduced chemical reprogramming cocktail could be achieved in this cell type (Aasen et al, 2008). Strikingly, keratinocytes treated with the 2c cocktail exhibited similar improvements in key aging markers as observed in fibroblasts and displayed no toxicity at the selected concentrations (Fig. EV1A,B). Notably, we observed a reduction in γH2AX levels, indicating decreased DNA damage, an

**Figure 3.  Optimized cocktail (2c) efficiently improves multiple molecular hallmarks of aging.**

(A) Immunofluorescence and quantification of γH2AX following TCP + Repsox (2c, 5 µM each) treatment (6 days, "6D"). (B, C) Immunofluorescence and quantification of H3K9me3 (B) and H3K27me3 (C) following 2c treatment (6D). (D) Senescence-associated beta-galactosidase (SA-beta-gal) staining and quantification following Doxorubicin (100 nM) in 2c pretreated fibroblasts (6D). (E) mRNA levels of senescence-associated *p21* expression following doxorubicin (100 nM) in 2c pretreated fibroblasts (29 days, "29D"). (F) SA-beta-gal staining and quantification of replicative-induced senescence (RIS) following long-term (29D) 2c treatment. (G) mRNA levels of senescence-associated and age-related stress response genes in the *p53* tumor suppressor pathway following 2c treatment. (H) MTS quantification of cell density following 2c treatment until confluence. (I) Fluorescence detection and quantification of ROS following 2c treatment (6D). Data were median ± IQR (A–C), mean ± SEM (D–I). (A–C, E–G, I) $n \geq 3$, (D, H) $n = 2$. Statistical significance was assessed by comparison to untreated control using paired two-tailed *t*-test (A–C, F, H, I), one-way ANOVA and Dunnett correction (D, E, G). OD optical density. Source data are available online for this figure.

increase in both H3K9me3 and H3K27me3 levels, reflecting enhanced heterochromatin integrity, and a significant decrease in SA-beta-gal levels (Fig. EV1C–F). Moreover, doxorubicin-induced stress in cells pretreated with 2c showed a significant decrease in both γH2AX and SA-beta-gal levels relative to doxorubicin-treated control, further underscoring the protective effects of 2c against genotoxic stress (Fig. EV1G,H). Overall, these results demonstrate that the reduced 2c cocktail effectively improves multiple age-related hallmarks, including genomic instability, epigenetic dysregulation, and cellular senescence, across different cell types, including both fibroblasts and keratinocytes.

In fibroblasts, immunofluorescence staining after one month following withdrawal of 2c treatment revealed increased levels of H3K9me3 and H3K27me3 relative to control, indicating a long-lasting positive effect upon treatment (Fig. EV2A,B). Furthermore, we performed a scratch assay in 2c-treated fibroblasts by introducing BJ cells (neonatal fibroblasts) as an additional control. Interestingly, we observed an initial increase in the rate of wound healing in the 2c-treated fibroblasts compared to the control, aligning with BJ fibroblasts, suggesting an improvement in regenerative capacity upon 2c treatment (Fig. EV2C).

Next, we performed bulk RNA sequencing on fibroblasts treated with either 2c or vehicle control. Similarly to 7c treatment, principal component analysis (PCA) showed that cells treated with 2c clustered separately from the control group indicating that a distinct transcriptomic profile emerges following this treatment (Fig. 4A). Moreover, expression and gene ontology (GO) enrichment analysis revealed that similarly to 7c, developmental processes were significantly upregulated following 2c treatment relative to control, alongside cell migration pathways, whereas RNA splicing, and post-translational events were significantly downregulated (Fig. 4B,C). In addition, we compared 7c and 2c transcriptomic profiles and found that DEGs were significantly overlapped, suggesting that associated transcriptional changes go in the same direction (Fig. 4D). Similarly, pathway analysis revealed significant pathways overlapped in the 2c and 7c groups, with many developmental pathways shared between both treatments (Fig. EV2D,E). Finally, gene signatures in 2c and 7c were inversely correlated to aging (Fig. EV2F).

In order to assess dedifferentiation upon treatment with chemical reprogramming cocktails, we performed immunostaining and transcriptional analysis of key fibroblast genes in 2c- and 7c-treated fibroblasts (Olova et al, 2019; Guan et al, 2022b). Supporting the induction of partial reprogramming, immunostaining revealed decreased expression of Collagen 1a in fibroblasts treated with 7c, whereas fibroblasts treated with 2c did not show a significant change relative to control (Fig. EV2G). RNA sequencing further supported these findings, revealing a global downregulation

of fibroblast markers in 7c-treated fibroblasts compared to controls, and to a lower extent upon 2c treatment (Fig. 4E). Taken together, these results indicate that the improvements in age-associated phenotypes observed upon partial chemical reprogramming treatment are dissociated from a loss of cellular identity, mirroring OSKM-induced reprogramming (Olova et al, 2019).

## 2c treatment increases *C. elegans* lifespan

Finally, in order to determine whether our 2c cocktail could also impact biological aging in vivo, we tested the effect of 2c treatment on the lifespan of a commonly used aging model organism, the nematode *Caenorhabditis elegans*. Towards this goal, we monitored survival in *C. elegans* treated with either 2c, Repsox, or TCP at three different concentrations (50, 100, or 200 µM) alongside a vehicle control. Strikingly, we observed that 2c treatment at 50 µM was sufficient to extend *C. elegans* median lifespan from 19 to 27 days, corresponding to a 42.1% increase relative to vehicle control (Table 1; Fig. 5A,B). To a lesser extent, Repsox or TCP alone at 50 µM also increased *C. elegans* median lifespan to 25 days, a 31.6% increase over vehicle control (Table 1; Fig. 5C,D). These results indicate that Repsox and TCP are each able to extend median lifespan in *C. elegans*, and when combined as part of the 2c cocktail, can lead to an even greater increase in median lifespan.

In addition, we observed dose-dependent effects across all treatments (Fig. 5B–D). Interestingly, the 2c cocktail or Repsox alone did not increase *C. elegans* lifespan at 200 µM (Fig. 5B–D). On the other hand, TCP still increased median lifespan by 15.8% at 100 or 200 µM relative to vehicle controls, even though it was most effective at 50 µM (Fig. 5D).

Next, we sought to determine the effect of 2c treatment to rescue age-associated changes in healthspan in *C. elegans*. Importantly, aging nematodes display a wide variety of healthspan declines with age, and similarly to mammalian species, including humans, undergo a strong loss in reproductive capacity. Notably, hermaphrodites display a peak progeny production on day 2 of adulthood, followed by a rapid decline during the first week of life. For this reason, total reproductive span, total progeny numbers, reproductive span to peak, and peak progeny numbers were recorded. In agreement with the literature, hermaphrodite nematodes displayed a peak progeny production on day 2 of adulthood and a total reproductive span of 5 days (Fig. 5E). Strikingly, treatment with 2c significantly increased total reproductive span to 7 days without affecting peak and total progeny numbers, highlighting the potential of 2c to enhance reproductive longevity without compromising reproductive output. On the other hand, treatment with the well-described lifespan and reproductive span extending drug Metformin extended lifespan, total reproductive span to a

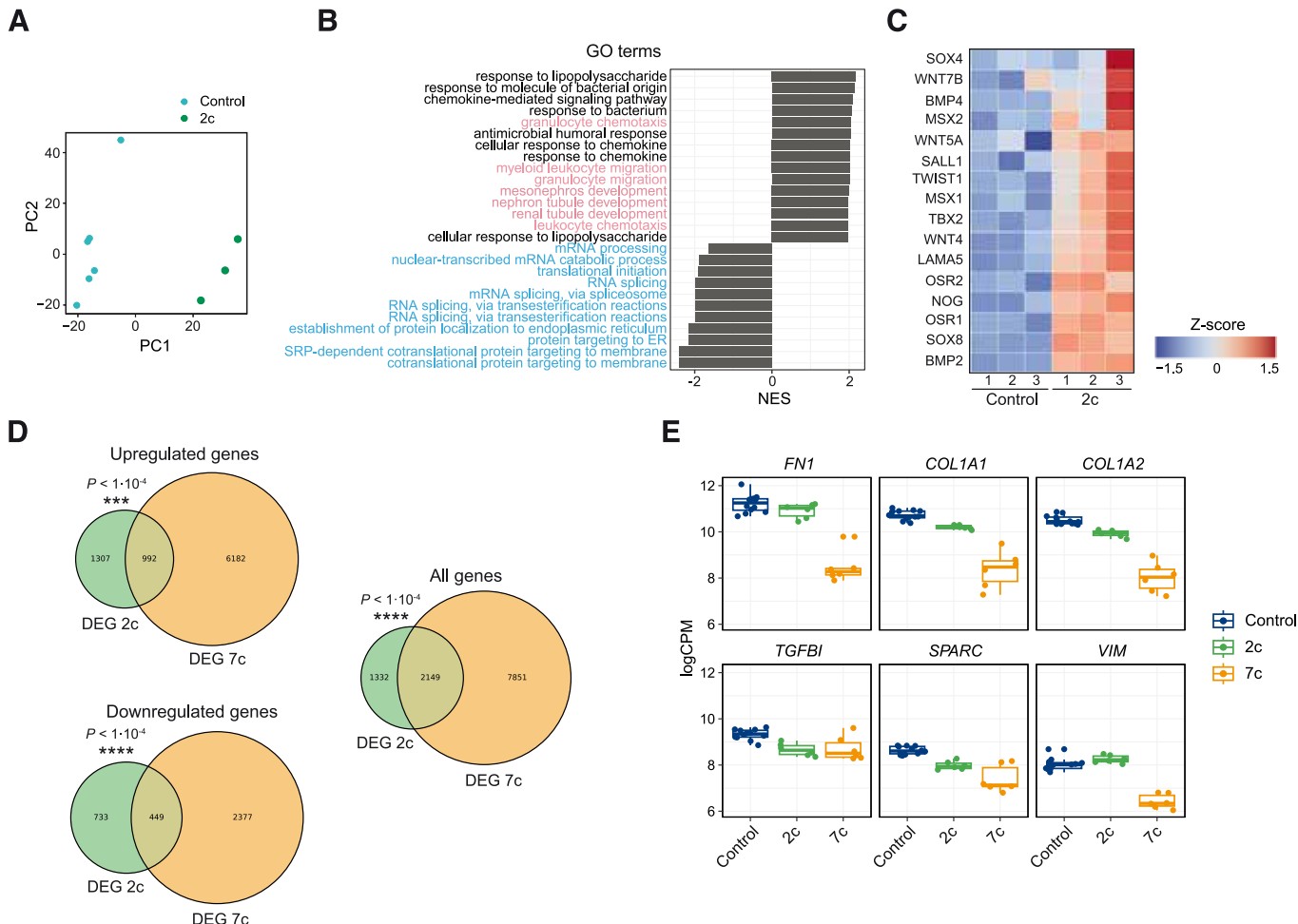

**Figure 4. Characterization of 2c transcriptomic profile.**

(A) Principal component analysis (PCA) of control (blue) and 2c-treated (green) fibroblasts. (B) Gene ontology (GO) enrichment analysis following 2c treatment with developmental and cell migration (in pink) and RNA splicing and post-translational (in blue) pathways highlighted. (C) Heatmap showing the expression pattern of differentially expressed genes associated with developmental pathways following 2c treatment. (D) Venn diagram showing the overlapping differentially expressed genes between 2c and 7c treatments. (E) Gene expression changes of key fibroblast markers following 2c and 7c treatment. Data are median ± IQR with whiskers extending to the furthest values within 1.5·IQR (E). (A–E) $n \geq 3$. Statistical significance was assessed by comparison to the untreated control using exact hypergeometric probability (total reference gene count of 18000) (D). NES normalized enrichment scores, CPM counts per million.

milder extent, and reduced peak and total progeny numbers, aligning with existing literature (Table 2; Fig. EV3A,B). Next, we examined the impact of 2c treatment on the senescent phenotype in *C. elegans*. Uterine tumors, a prominent form of senescent pathology in nematodes, arise from unfertilized oocytes that enter the uterus and become hypertrophic. For this reason, we decided to analyze the development and progression of these complex masses, which can eventually expand and occupy much of the animal's body cavity with age (Wang et al, 2018; Kern et al, 2023). Notably, we found a significant reduction in tumor size in 2c-treated animals at old age (Figs. 5F and EV3C). Finally, we observed that the age-related decrease in body size was slowed down upon 2c treatment in *C. elegans* (Fig. EV3D).

One of the most informative metrics of health in *C. elegans* is motor functions; we tracked crawling speed over time and assessed the effect of 2c on this phenotype. In agreement with the literature, we observed a constant reduction in movement speed with age

(Figs. 5G; EV3E). In addition, 2c treatment was able to significantly increase average movement speed consistently throughout *C. elegans* lifespan, while also increasing maximum speed (Figs. 5G and EV3E,F). Furthermore, thrashing analysis in liquid media revealed similar healthspan benefits upon 2c treatment, with swimming speed and activity being significantly improved in old worms (Fig. 5H,I). In addition, although to a lesser extent, multiple locomotor parameters, including swimming asymmetry, dynamic amplitude, as well as maximum speed, showed a tendency towards improvements in the 2c-treated group (Fig. EV3G).

Next, since oxidative stress as a consequence of mitochondrial dysfunction is implicated in the pathogenesis of many age-related diseases, we assessed *C. elegans* sensitivity to oxidative stress following exposure to paraquat. Remarkably, we observed an increase in survival and resistance of worms treated with 2c upon exposure to the ROS-generating compound paraquat (Fig. 5J). Notably, exposure to 40 mM paraquat reduced the median lifespan of middle-aged *C. elegans* to

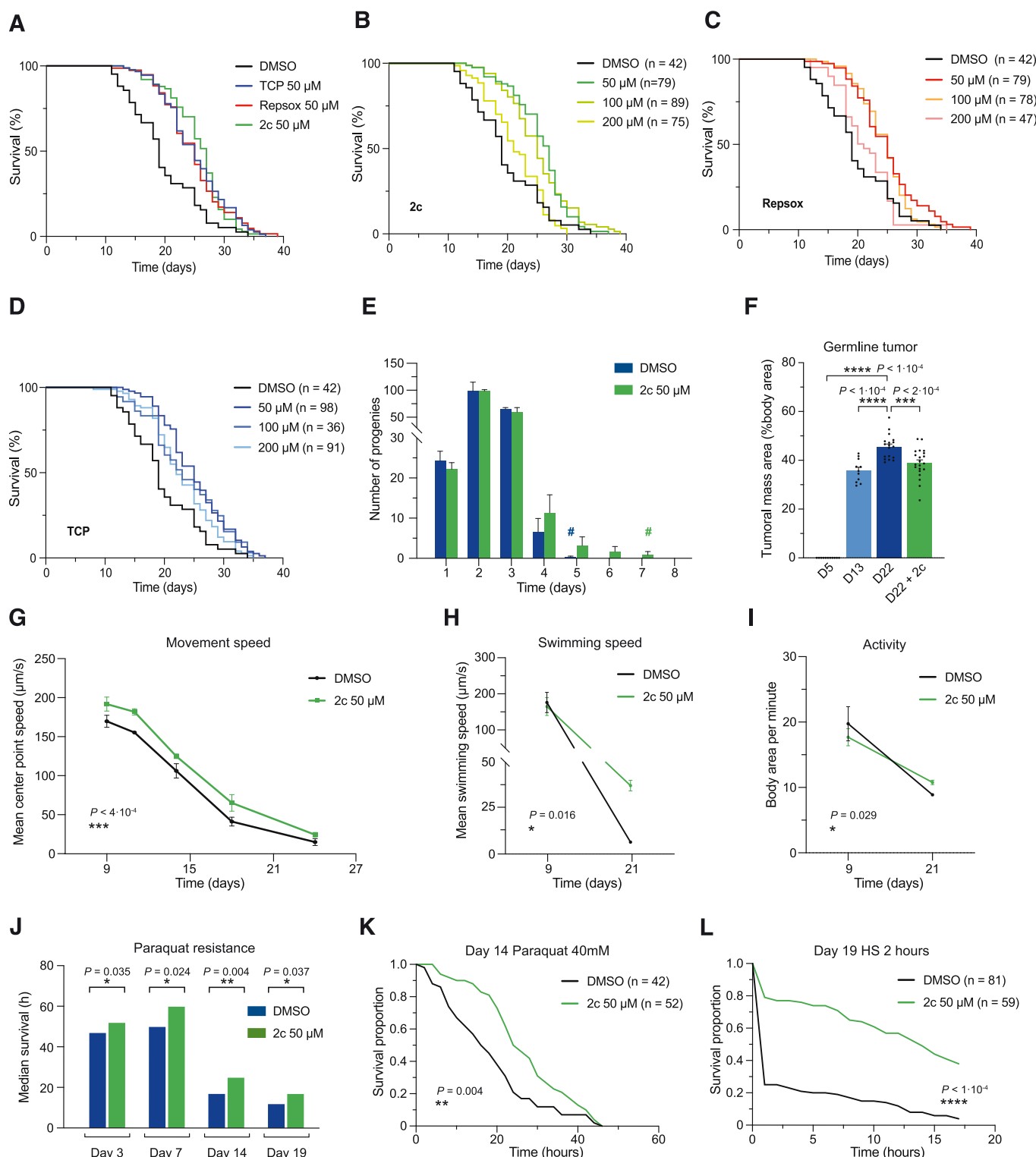

17 h, while 2c treatment increased median survival to 25 h, a 47.1% increase over vehicle control (Fig. 5J,K). Finally, we conducted thermotolerance assays as final age-associated markers of health. As expected, acute heat stress at 37 °C significantly impaired *C. elegans* survival in both young and old nematodes. Impressively, we found that *C. elegans* treated with 2c had a significantly higher resistance to heat stress, confirming the capacity of the reduced two-chemical cocktail to improve markers of health in *C. elegans* (Figs. 5L and EV3H).

Taken together, these data demonstrate that the optimized 2c cocktail can both ameliorate multiple aging hallmarks in aged human fibroblasts in vitro and extend both *C. elegans* lifespan and healthspan in vivo.

**Figure 5.   Treatment with 2c increases *C. elegans* lifespan and healthspan.**

(A) Survival of N2 *C. elegans* upon treatment with TCP (50 μM), Repsox (50 μM), and 2c (TCP + Repsox, 50 μM each). (B–D) Survival of N2 *C. elegans* upon treatment with 2c (B), Repsox (C), and TCP (D) at 50, 100, or 200 μM. (E) Progeny generation and reproductive span (indicated with #) of the unmated hermaphrodite *C. elegans* upon treatment with 2c. (F) Quantification of the germline tumoral mass development at young (day 5), middle (day 13), and old age (day 22) upon 2c treatment. (G, H) Mean movement speed (G), swimming speed (H), and swimming activity (I) of N2 *C. elegans* upon treatment with 2c. (J) Resistance to oxidative stress was measured at 3, 7, 14, and 19 days of adulthood upon 2c treatment. (K) Survival of *C. elegans* to paraquat-induced oxidative stress upon 2c treatment at 14 days of adulthood. (L) Thermotolerance of *C. elegans* to 2 h HS at day 19 of adulthood upon 2c treatment. Data were mean ± SEM (E–I). (E–G, J) $n \geq 10$, (H–I) $n \geq 4$. The number of animals and measures are detailed in the Methods. Statistical significance was assessed by comparison to vehicle control using log-rank (Mantel-Cox) test for each lifespan (A–D, J–L), one-way ANOVA and Dunnett correction (F), two-way ANOVA mixed-effects analysis and Geisser-greenhouse correction (G), paired two-tailed *t*-test for day 21 of adulthood (H, I). HS Heat shock. Source data are available online for this figure.

**Table 1.   Summary of lifespan extensions upon 2c treatment in *C. elegans*.**

| Drug | Dose (μM) | Median lifespan (d) | Median lifespan increase (%) | Maximum lifespan (d) | *n* | Log-rank *p* value |
|---|---|---|---|---|---|---|
| DMSO (control) | | 19 | | 27 | 42 | |
| TCP | 50 | 25 | 31.6 | 32 | 98 | $<1 \cdot 10^{-4}$ |
| | 100 | 23 | 21.1 | 32 | 36 | 0.0077 |
| | 200 | 22 | 15.8 | 30 | 91 | 0.0106 |
| Repsox | 50 | 25 | 31.6 | 32 | 79 | $<1 \cdot 10^{-4}$ |
| | 100 | 25 | 31.6 | 29 | 78 | 0.0013 |
| | 200 | 21 | 10.5 | 26 | 47 | *n.s.* (0.479) |
| 2c | 50 | 27 | 42.1 | 30 | 79 | $<1 \cdot 10^{-5}$ |
| | 100 | 25 | 31.6 | 32 | 89 | $<1 \cdot 10^{-4}$ |
| | 200 | 21 | 10.5 | 27 | 75 | *n.s.* (0.447) |

Summary of survival assay results including median lifespan, maximal (90%) lifespan, and statistical analysis. Median lifespan increases relative to vehicle control. Statistical significance was assessed by comparison to vehicle control using the log-rank (Mantel-Cox) test.

## Discussion

The molecular identity and age of somatic cells have proven to be plastic states that can be reset by cellular reprogramming (Gurdon, 1962; Campbell et al, 1996; Takahashi and Yamanaka, 2006; Lapasset et al, 2011). As aging and age-associated diseases are a major societal burden, the need for aging interventions such as cellular reprogramming has grown (Mahmoudi et al, 2019; Garmany et al, 2021). Although multiple groups have now demonstrated that in vivo partial reprogramming via transient application of OSKM can rejuvenate molecular hallmarks of aging, restore tissue function, and extend lifespan in mouse models, the risks of oncogenesis and inefficient gene delivery hinders clinical development (Ocampo et al, 2016b; Kurita et al, 2018; de Lázaro et al, 2019; Lu et al, 2020; Rodríguez-Matellán et al, 2020; Sarkar et al, 2020; Chen et al, 2021; Cheng et al, 2022; Hishida et al, 2022). Interestingly, a more translational approach for the induction of cellular reprogramming based on the use of small molecules has been recently developed (Hou et al, 2013; Zhao et al, 2015; Cao et al, 2018; Guan et al, 2022a). Still, the effects of small molecule-induced cellular reprogramming on aging hallmarks and lifespan were, until now, unknown.

Here, we demonstrated for the first time that partial chemical reprogramming induces multiparameter rejuvenation of key aging hallmarks, including genomic instability, epigenetic dysregulation, and cellular senescence in vitro while simultaneously extending the lifespan of *C. elegans* in vivo. In particular, we demonstrated that the seven-chemical reprogramming cocktail, defined by Hou et al, was able to improve age-associated DNA damage, epigenetic alterations, and induce a unique transcriptomic profile enriched for developmental processes in aged human fibroblasts in vitro. Our observations that 7c significantly impaired proliferation and increased ROS levels might contribute to the previously observed low efficiency of mouse iPSC induction (Hou et al, 2013). The upregulation of ROS levels and impaired proliferation observed upon 7c treatment could be explained by cytostaticity or cytotoxicity arising from some of the single small molecules included in the cocktail (Appendix Fig. S2). Recently, Guan et al, 2022a described the use of chemical reprogramming of human somatic cells to iPSC employing a combination of small molecules derived from the original mouse chemical cocktail (Hou et al, 2013). As part of their new cocktail, they identified JNKIN8 as a key small molecule for the acquisition of the plasticity signature in human cells and highlighted the roles of *MSX1* and *MSX2* in the intermediate plastic state during induction to pluripotency. Intriguingly, overexpression of both regulators was observed upon 7c treatment, potentially implying that the 7c cocktail is not only successful in initiating reprogramming, but also in guiding cells through critical intermediate states.

We further revealed that an optimized two-compound cocktail (2c) is sufficient to decrease the levels of the DNA damage marker γH2AX, increase H3K9me3 and H3K27me3, prevent both replicative and genotoxic induced senescence, and decrease SASP and oxidative stress. In addition, application of this treatment to human epidermal keratinocytes revealed similar benefits, suggesting that the effects of the 2c cocktail are not cell-type specific but rather broadly applicable across different cell lineages in humans. These results are supported by a recent study from our group (Paine et al, 2024), showing that treatment with Repsox alone, through inhibition of TGFβ, was able to decrease levels of γH2AX and restore the DNA methylation clock in the fibroblasts from the premature aging mouse model Ercc1. On the other hand, TCP is an inhibitor of the histone demethylase LSD1, as previously shown by Clark et al, that can increase H3K9me3 levels, normally decreased during aging in human tissues. Most importantly, we found that our 2c cocktail applied in vivo was able to extend the median lifespan of *C. elegans* by 42.1%. Interestingly, the highest doses of 2c or single compounds were less effective in extending lifespan,

**Table 2. Summary of reproductive aging upon two lifespan-extending interventions in *C. elegans*.**

| Drug | Dose (µM) | Longitudinal experiment | Total reproductive span (days) | Total progeny number | Reproductive span to peak (days) | Peak progeny number | Log-rank p value |
|---|---|---|---|---|---|---|---|
| DMSO (control) | | yes | 5 | 195 | 2 | 101 | |
| 2c | 50 | yes | 7 | 198 | 2 | 99 | $<1\cdot10^{-4}$ |
| Metformin | 50,000 | yes | 6 | 182 | 2 | 84 | 0.0283 |

Summary of reproductive aging assay results upon treatment with 2c and Metformin. Experiments were performed longitudinally and total reproductive span (time from adult day 0 until last day of progeny production), total progeny numbers (total number of progeny produced), reproductive span to peak (time from adult day 0 until peak day of progeny production), and peak progeny production (number of progeny produced on day of peak progeny production) were recorded. The number of animals and measures are detailed in the Methods. Statistical significance was assessed by comparison to vehicle control using an unpaired two-tailed *t*-test of total reproductive span.

indicating potential off-target effects that may require further optimization.

Previous reports have demonstrated that in vivo treatment with a modified small-molecule reprogramming cocktail similar to Hou et al, could enhance regeneration in the liver and heart, thus providing proof-of-principle that treatment with these reprogramming-associated chemicals could benefit tissue repair and, more recently, enable autologous islet replacement in type 1 diabetes (Tang and Cheng, 2017; Huang et al, 2018; Wang et al, 2024). On the other hand, we have now shown that chemical reprogramming can improve multiple molecular hallmarks of aging similarly to OSKM-induced reprogramming and can extend both the lifespan and the healthspan of *C. elegans*. Multiparameter rejuvenation across aging hallmarks is a defining trait of cellular reprogramming, albeit future work is required to properly identify the mechanisms responsible for these benefits (Chondronasiou et al, 2022; Gill et al, 2022). In this line, attempts to induce multiparameter amelioration are now emerging as a strategy for rejuvenation even outside the field of cellular reprogramming. In this regard, Shaposhnikov et al, recently demonstrated a synergistic effect by targeting multiple aging features simultaneously, producing a significant increase in *Drosophila melanogaster* lifespan compared to single interventions (Shaposhnikov et al, 2022).

Similarly, recent single-factor work showed that overexpressing the splicing regulator SRSF1 or a newly discovered single-gene modulator alone can recapitulate broad rejuvenation. These interventions achieve OSKM-level efficacy while preserving fibroblast-specific transcriptomic signatures and producing no pluripotent colonies, thus decoupling pluripotency from cell rejuvenation, an outcome our small-molecule approach likewise mirrors (Plesa et al, 2023; Camillo et al, 2025; Mitchell et al, 2024).

Importantly, several translational advantages highlight the potential use of chemical-induced partial reprogramming for the amelioration of age-associated phenotypes, including the fact that small molecules are cell-permeable and therefore easy to deliver. Furthermore, their effects can be modulated via dosage, and are transient and reversible, thus avoiding oncogenic pitfalls associated with transcription factor induction (Zhao, 2019). In this proof-of-principle study, our observations indicate that chemical reprogramming represents both a valuable opportunity for the development of future anti-aging interventions, along with the mechanistic understanding of the complex inter-relationships of aging hallmarks and their respective amelioration.

# Methods

### Reagents and tools table

| Reagent/resource | Reference or source | Identifier or catalog number |
|---|---|---|
| **Experimental models** | | |
| *C. elegans* | CGC | N2 |
| Human dermal fibroblasts | Biopredic | |
| Human epidermal keratinocytes | Biopredic | |
| **Recombinant DNA** | | |
| **Antibodies** | | |
| anti-H3K27me3 | Abcam | ab192985 |
| anti-H3K9me3 | Cell Signaling | 13969 |
| anti-γH2AX | Cell Signaling | 9718 |
| anti-Ki67 | Cell Signaling | 15580 |
| anti-COL1A1 | Cell Signaling | 260043 |
| anti-H3 | Bioconcept | 13969 |
| anti-β-actin | Sigma | A2228 |
| anti-Rabbit | Thermo Fisher | A32790 |
| anti-rabbit immunoglobulins/HRP | Agilent | P0448 |
| anti-mouse immunoglobulins/HRP | Agilent | P0447 |
| DAPI | Roth | 6843.1 |
| **Oligonucleotides and other sequence-based reagents** | | |
| **Chemicals, enzymes and other reagents** | | |
| Collagenase I | Sigma | C0130 |
| Dispase II | Sigma | D4693 |
| DMEM | Gibco | 11960085 |
| Non-essential amino acids | Gibco | 11140035 |
| GlutaMax | Gibco | 35050061 |
| Sodium pyruvate | Gibco | 11360039 |
| 10% fetal bovine serum (FBS) | Hyclone | SH30088.03 |
| CnT-prime epithelial proliferation medium | CELLnTEC | CnT-PR |
| 4% paraformaldehyde | Roth | 0964.1 |
| 1% bovine serum albumin | Sigma | A9647-50G |
| PBST (0.2% Triton X-100 in PBS) | Roth | 3051.3 |

| Reagent/resource | Reference or source | Identifier or catalog number |
| --- | --- | --- |
| Fluoromount-G | Thermo Fisher | 00-4958-02 |
| DHE | Thermo Fisher | D11347 |
| Paraquat | Thermo Fisher | 856177 |
| Valproic Acid | Cayman | 13033 |
| CHIR99021 | Cayman | 13122 |
| Repsox | Cayman | 14794 |
| Forskolin | Cayman | 11018 |
| Doxorubicin | Cayman | 15007 |
| TCP | Acros Organics | 130472500 |
| DZNep | APExBIO | A8182 |
| Metformin | APExBIO | B1970 |
| TTNPB | Seleckchem | S4627 |
| X-beta-Gal | Roth | 2315.3 |
| DNase treatment | Qiagen | 79254 |
| iScript™ gDNA Clear cDNA Synthesis | Bio-Rad | 1725035 BUN |
| SsoAdvanced SYBR Green Supermix | Bio-Rad | 1725272 |
| CellTiter 96® AQueous One Solution | Promega | G3580 |
| Qubit RNA BR Assay Kit | Thermo Fisher | Q10211 |
| Monarch Total RNA Miniprep Kit | New England Biolabs | T2010S |
| PBS | Gibco | 21600069 |
| crystal violet | Roth | T123.2 |
| Software | | |
| R | | |
| STAR aligner (v2.7.9a) | | |
| FastQC | | |
| featureCounts (subread) | | |
| AnnotationDbi, limma, clusterProfiler | | |
| IncuCyte 2019B Rev2 software | Essen Bioscience | |
| GraphPad Prism 9.0.0 | | |
| SPSS 27.0.1.0 | IBM® SPSS® Statistics | |
| Fiji | | |
| Other | | |
| Ti2 Spinning Disk | Nikon | Yokogawa CSU-W1 |
| NovaSeq 6000 | Illumina | |
| QuantStudio™ real-time PCR instrument | Thermo Fisher | 12 K Flex |
| 384-well PCR plates | Thermo Fisher | AB1384 |
| Thermocycler | Bio-Rad | 1861086 |
| BioTek Epoch 2 | | |
| 96-well plates | Essen Bioscience | 4379 |
| IncuCyte S3 | Essen Bioscience | |

## Cell culture, maintenance, and treatment

Human dermal fibroblasts were freshly extracted using Collagenase I (Sigma, C0130) and Dispase II (Sigma, D4693) and cultured in DMEM (Gibco, 11960085) containing non-essential amino acids (Gibco, 11140035), GlutaMax (Gibco, 35050061), sodium pyruvate (Gibco, 11360039), and 10% fetal bovine serum (FBS, Hyclone, SH30088.03) at 37 °C in hypoxic conditions (3% $O_2$). Human epidermal keratinocytes were freshly extracted using Dispase II (Sigma, D4693) and cultured in CnT-Prime epithelial proliferation medium (CELLnTEC, CnT-PR). Subsequently, fibroblasts and keratinocytes were passaged and cultured according to standard protocols. Aged donor samples were of 56, 70, and 83 years of age for fibroblasts, and 25, 26, 33, and 53 years of age for keratinocytes. All samples were tested for Hepatitis BC, HIV1, and HIV2 (carried out on the patient by serological screening). Cells were cultured in vitro and discarded when reaching passage 15, corresponding to population doubling (PD) 28–30. Two fibroblast lines (56 y and 83 y) were utilized with three repetitions for each experiment when indicated. Additionally, for the supplementary 2c experiments, three fibroblast lines (56 y, 70 y, and 83 y) and four keratinocyte lines (25 y, 26 y, 33 y, and 53 y) were used to perform the experiments. The decision to treat cells with 7c or 2c for 6 days was based on observations indicating early enhancements in H3K9me3 as early as two days post-treatment, while DNA damage levels were only consistently reduced upon 6 days of chemical-induced partial reprogramming with 7c. In addition, a shorter treatment period aimed to minimize stress on cells by reducing the need for multiple passages. Overall, the 6-day duration struck a balance between achieving improvements in age-associated phenotypes while maintaining cellular identity.

## Ethics and consent

Biological materials used in this study were obtained from Biopredic (France) with appropriate ethical approval. Informed consent was obtained from all donors prior to sample collection. All procedures were conducted in accordance with the ethical standards of the institutional and/or national research committee, the 1964 Declaration of Helsinki and its later amendments, and the principles outlined in the US Department of Health and Human Services Belmont Report.

## Immunofluorescence staining

Cells were washed with fresh PBS and then fixed with 4% paraformaldehyde (Roth, 0964.1) in PBS at room temperature (RT) for 15 min. After fixation, cells were washed three times, followed by a blocking and permeabilization step in 1% bovine serum albumin (Sigma, A9647-50G) in PBST (0.2% Triton X-100 in PBS) for 60 min (Roth, 3051.3). Cells were then incubated at 4 °C overnight with the appropriate primary antibody, washed in PBS, followed by secondary antibody incubation with DAPI staining at RT for 60 min. Coverslips were mounted using Fluoromount-G (Thermo Fisher, 00-4958-02), dried at RT in the dark for several hours, stored at 4 °C until ready to image and −20 °C for long-term. Future analyses might include additional markers of DNA damage, such as 53BP1 immunostaining, to complement our current methodology and provide a more comprehensive assessment of DNA damage.

## Immunofluorescence imaging

Confocal image acquisition was performed using the Ti2 Yokogawa CSU-W1 Spinning Disk (Nikon), using the 100X objective and with 15 z-sections of 0.3 μm intervals. Appropriate lasers were used (405 nm and 488 nm) with a typical laser intensity set to 5–10% transmission of the maximum intensity for methylated histones, Ki67 and ROS, and 30–40% for phosphorylated histones. Exposure time and binning were established separately to assure avoidance of signal saturation.

## Antibodies and compounds

Antibodies were provided by the following companies. Abcam: anti-H3K27me3 (ab192985, 1:400); Cell Signaling: anti-H3K9me3 (13969, 1:400), anti-γH2AX (9718, 1:400), anti-Ki67 (15580, 1:400), anti-COL1A1 (260043, 1:800); Thermofisher: anti-Rabbit (A32790, 1:800); Roth: DAPI (6843.1, 500 ng/mL)

Compounds were purchased from the following companies. Thermo Fisher: DHE (D11347), Paraquat (856177); Cayman: Valproic Acid (13033), CHIR99021 (13122), Repsox (14794), Forskolin (11018), Doxorubicin (15007); Acros Organics: TCP (130472500); APExBIO: DZNep (A8182), Metformin (B1970); Seleckchem: TTNPB (S4627); Roth: X-beta-Gal (2315.3). Concentrations in Appendix Table S1.

## RNA analysis

Total RNA was extracted using Monarch Total RNA Miniprep Kit (New England Biolabs, T2010S) according to the manufacturer's instructions with DNase treatment (Qiagen, 79254) for 15 min (1:8 in DNase buffer). Total RNA concentrations were determined using the Qubit RNA BR Assay Kit (Thermo Fisher, Q10211). cDNA synthesis was performed by adding 4 μL of iScript™ gDNA Clear cDNA Synthesis (Bio-Rad, 1725035BUN) to 500 ng of RNA sample and run in a Thermocycler (Bio-Rad, 1861086) with the following protocol: 5 min at 25 °C for priming, 20 min at 46 °C for reverse transcription, and 1 min at 95 °C for enzyme inactivation. Final cDNA was diluted 1:5 using autoclaved water and stored at −20 °C. qRT-PCR was performed using SsoAdvanced SYBR Green Supermix (Bio-Rad, 1725272) in 384-well PCR plates (Thermo Fisher, AB1384) using the QuantStudio™ 12 K Flex Real-time PCR System instrument (Thermofisher). Forward and reverse primers (1:1) were used at a final concentration of 5 μM with 1 μL of cDNA sample. Primer sequences are listed in Appendix Table S2.

## RNA sequencing, processing, and analysis

RNA-Seq library preparation and sequencing was performed by Novogene (UK) Company Limited on an Illumina NovaSeq 6000 in 150 bp paired-end mode. Raw FASTQ files were assessed for quality, adapter content and duplication rates with FastQC. Reads were aligned to the Human genome (GRCh38) using the STAR aligner (v2.7.9a) with '--sjdbOverhang 100' (Dobin et al, 2013). The number of reads per gene was quantified using the featureCounts function in the subread package (Liao et al, 2013). Ensembl transcripts were mapped to gene symbols using the mapIds function in the AnnotationDbi package (Pagès et al, 2022) with

the org.Hs.eg.db reference package (Carlson, 2019). Raw counts were normalized by library size and converted to counts per million (CPM) for downstream analysis. Dimensionality reduction was performed via Principal Component Analysis (PCA) using the R software. Differentially expressed genes (DEG) were computed by the limma R package (Ritchie et al, 2015), by fitting a linear model on each gene, with an adjusted $p$ value of 0.05. Gene set enrichment analysis (GSEA) for gene ontology (GO)(Ashburner et al, 2000) was performed using the clusterProfiler package (Wu et al, 2021) from the list of DEG (with a valid Entrez ID) ranked by logFoldChange. We used org.HS.eg.db as a reference, selected Biological Processes (BP) only, and an adjusted $p$ value of 0.05 (Bonferroni). Pathways were ranked by normalized enrichment score (NES). Z-score were calculated for each gene to plot as a heatmap.

## MTS cell proliferation assay

Cell viability and proliferation assays were performed by the tetrazolium MTS assay. Control and treated cells were cultured for 1 day in 96-well plates, then treated with small molecules for 3 consecutive days before incubation with 120 μL fresh media containing 20 μL of CellTiter 96® AQueous One Solution (Promega, G3580) for 1 to 4 h at 37 °C in a humidified, 5% $CO_2$ atmosphere. The amount of product formed was measured by recording the absorbance at 490 nm using a BioTek Epoch 2 microplate reader. The relative proportion of viable cells was determined as a relative reduction of the optical density (OD) compared to the control OD.

## Crystal violet staining

Cells were cultured as previously described for the MTS cell proliferation assay. Culture media was then carefully removed from wells and cells were washed three consecutive times with room temperature PBS (Gibco, 21600069) followed by 45 min incubation with crystal violet (Roth, T123.2) solution (Crystal Violet 0.05%, Formaldehyde 0.4%, Methanol 1% in PBS 1X). Plates were then washed by careful immersion in tap water two times, drained upside down, and air dried. Finally, solubilization in 1% SDS was performed on an orbital shaker until no dense areas of coloration persisted, and absorbance was measured at 570 nm using a BioTek Epoch 2 microplate reader. The relative proportion of viable cells was determined as a relative reduction of the optical density (OD) compared to the control OD.

## Senescence-associated β-galactosidase assay

Senescence-associated beta-galactosidase (SA-βgal) assay was performed as described in the literature (Debacq-Chainiaux et al, 2009). Briefly, light fixation was performed on cells plated on glass coverslips using a solution of 3% paraformaldehyde and 0.2% glutaraldehyde in PBS buffer for 5 min. Fixation solution was then removed, wells were washed several times and stained overnight at 37 °C in a CO2-free incubator in a solution of 40 mM citric acid/Na phosphate buffer, 5 mM K4[Fe (CN)6]3H2O, 5 mM K3[Fe (CN)6], 150 mM sodium chloride, 2 mM magnesium chloride, and 1 mg/mL X-gal (Roth, 2315.1) with a pH of 5.9–6.0. Finally, coverslips were

stained with DAPI, followed by a standard immunofluorescence protocol, images were taken using bright-field microscopy, and the proportion of β-Gal-positive cells was then quantified.

## Reactive oxygen species assay

Mitochondrial reactive oxygen species (ROS) were measured using the superoxide indicator dihydroethidium (DHE). Briefly, first, cells were incubated in fresh FBS-free media containing 5 μM DHE and incubated at 37 °C in a humidified, 5% $CO_2$ atmosphere for 30 min. Following incubation, wells were washed with room temperature PBS, fixed with 4% paraformaldehyde for 15 min and then stained with DAPI. Immediately, images were taken at 554 nm and the standard immunofluorescence imaging protocol was followed.

## Scratch assay

Cells were cultured as previously described for the MTS cell proliferation assay. Wound was created using the Woundmaker Tool (Incucyte) on 90% confluent cultures cultivated in 96-well plates (Essen Bioscience, 4379). Images were taken every 4 h for 48 h in the IncuCyte S3 (Essen Bioscience) and analyses were carried out using the IncuCyte 2019B Rev2 software (Essen Bioscience).

## Quantification and statistical methods

The analysis of immunofluorescence microscopy images was performed using ImageJ 2.1.0. A minimum of 50–100 cells were imaged per condition. Maximal projections of z-stacks were analyzed, and total fluorescence intensity per cell was determined.

All statistical parameters, such as statistical analysis, statistical significance, and n values, are reported in the figure legends. IQR Interquartile Range, SEM Standard Error of Mean. Measurements were taken from distinct samples. Sample sizes were selected based on standard practices to ensure biological relevance and reproducibility. No formal statistical methods were used to predetermine sample size. Microscopy data acquisition and analysis were performed using semi-automated pipelines, ensuring objective and blinded quantification. No explicit inclusion or exclusion criteria were pre-established; all samples meeting technical quality standards were included. No data points were omitted unless due to technical failure, which was documented and justified. For figures involving statistical comparisons, appropriate tests were selected based on data distribution and group variance, which were visually assessed when relevant. Variation within groups is indicated by error bars, and variance between groups was assumed to be similar unless otherwise noted. Statistical analyses were performed using GraphPad Prism 9.0.0. For histone methylation quantification, outliers were systematically removed using the ROUT method (Q = 1%), and normality was assumed when applicable. For in vivo experiments, n corresponds to the number of animals and statistical analysis was performed using SPSS 27.0.1.0 (IBM® SPSS® Statistics).

## Statistical methods

Sample sizes were selected based on standard practices to ensure biological relevance and reproducibility. No formal statistical methods were used to predetermine sample size. Allocation of

samples was not randomized, and no blinding was applied during experimental procedures; however, microscopy data were analyzed using automated pipelines in FIJI, ensuring objective and blinded quantification. No explicit inclusion or exclusion criteria were pre-established; all samples meeting technical quality standards were included. No data points were omitted unless due to technical failure, which was documented and justified. For figures involving statistical comparisons, appropriate tests were selected based on data distribution and group variance, which were visually assessed when relevant. Variation within groups is indicated by error bars, and variance between groups was assumed to be similar unless otherwise noted.

## C. elegans strains and maintenance

Wild-type C. elegans (N2) were obtained from the Caenorhabditis Genetics Center (CGC), University of Minnesota, USA. N2 wildtype worms were maintained at 20 °C and were grown on standard Nematode Growth Media (NGM) plates.

## Lifespan analysis

Animals were synchronized and lifespan analyses were conducted at 20 °C as previously described (Porta-de-la-Riva et al, 2012) and were transferred onto NGM plates containing treatment or vehicle at stage L4. TCP was dissolved in water at 100 mM, Repsox was dissolved in DMSO at 200 mM, and Metformin was dissolved in water at 1 M. TCP and Repsox were added directly into the molten agar to a final concentration of 50, 100, or 200 μM each before pouring. Metformin was added directly into the molten agar to a final concentration of 50 mM before pouring. After proper drying of plates, UV-killed OP50 bacteria were seeded (150 μl of 120 mg/ml UV-killed OP50 per P60 plate) and FUDR (150 μM) added as a reproductive suppressant. Treated and control plates contained an equivalent DMSO concentration. Animals that crawled off the plate or displayed extruded internal organs were censored. Lifespan analyses were assessed manually by counting live and dead animals based on movement.

## Reproductive span assays

Animals were synchronized, transferred onto NGM control and treated plates at stage L4, and reproductive span analyses were conducted at 20 °C. ≥10 worms per treatment were plated in groups of 2–4 worms and transferred daily into fresh drug-free and 2c NGM plates. Images were captured using a Nikon Stereo Microscope 24 and 48 h after transferring the worms to assess the number of viable progenies. The average number of progenies per worm per day was then determined. Progeny counts were performed using ImageJ 2.1.0, and the last day of progeny production was identified as the day preceding the first day on which no viable progenies were observed. Experiments were performed longitudinally and total reproductive span (time from adult day 0 until last day of progeny production), total progeny numbers (total number of progenies produced), reproductive span to peak (time from adult day 0 until peak day of progeny production), and peak progeny production (number of progenies produced on day of peak progeny production) were recorded as previously described (Scharf et al, 2021).

**The paper explained**

**Problem**

As aging and age-related diseases place an increasing burden on society, there is an urgent need to develop effective and innovative strategies to mitigate their impact. Cellular reprogramming has emerged as a promising approach to target aging at its root, capable of reversing molecular and functional hallmarks of aging while extending both lifespan and healthspan. However, current reprogramming methods rely on genetic manipulation, limiting the translational potential and raising significant safety concerns, including the risk of tumorigenesis and loss of cellular function.

**Results**

Here, we demonstrate for the first time that partial chemical reprogramming using a seven-compound cocktail (7c) can induce multi-parametric rejuvenation across key hallmarks of aging. We further identified an optimized two-compound cocktail (2c) that is sufficient to improve additional aging phenotypes in vitro. Finally, application of this reduced reprogramming cocktail significantly improved multiple markers of aging, stress-resistance, and healthspan in *C. elegans*, leading to a median lifespan extension of over 42% in vivo.

**Impact**

This work establishes a novel chemical approach to partial cellular reprogramming, offering a clinically viable alternative to genetic methods. By demonstrating conserved rejuvenation effects in both human cells and a whole-organism model, our findings provide a foundation for the development of a novel pharmacological intervention that targets aging at its root.

## Locomotor assays

Animals were synchronized, transferred onto NGM control and treated plates at stage L4, and healthspan analyses were conducted at 20 °C. For movement assays in solid media, ≥10 worms per treatment per age were transferred onto fresh drug-free NGM plates at 9, 11, 14, 18, and 24 days of adulthood and recorded two to six times for 60 s. For swimming assays in liquid media, four to six worms per treatment per age were transferred into drug-free M9 buffer at 9 and 21 days of adulthood and recorded for 60 s. Images of ≥10 worms per treatment were used to determine the mean body length. Analysis were carried out using the CeleST computer vision software (Restif et al, 2014).

## Paraquat assays

Animals were synchronized, transferred onto NGM control and treated plates at stage L4, and healthspan analyses were conducted at 20 °C. At the desired age, similarly to standard CITP procedure (Lucanic et al, 2017; Banse et al, 2022, 2023), worms were added to plates containing 40 mM methyl viologen dichloride hydrate (Paraquat) and lifespan analyses were assessed manually at room temperature by counting live and dead animals based on movement.

## Thermotolerance assays

Animal and plate preparations were identical to paraquat assays. At the desired age, worms were transferred onto pre-heated drug-free NGM plates for 2 or 4 h at 37 °C. Following this acute heat stress, worms were transferred once again onto fresh room temperature (RT) NGM plates. Lifespans were then assessed manually at RT by counting live and dead animals based on movement.

## Data availability

The raw data (FASTQ) and count data (logCPM) for the RNA-seq profiling of human dermal fibroblasts treated with 2c and 7c have been deposited in the Gene Expression Omnibus database (https://www.ncbi.nlm.nih.gov/geo/) under accession number GSE297984 (https://www.ncbi.nlm.nih.gov/geo/query/acc.cgi?acc=GSE297984). Imaging data are available in the BioStudies database (https://www.ebi.ac.uk/biostudies/) under accession number S-BIAD2006 (https://www.ebi.ac.uk/biostudies/bioimages/studies/S-BIAD2006).

The source data of this paper are collected in the following database record: biostudies:S-SCDT-10_1038-S44321-025-00265-9.

## Peer review information

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

## Acknowledgements

We would like to thank all members of the Ocampo laboratory for helpful discussions. In addition, we would like to thank the UNIL Cellular Imaging Facility and especially Luigi Bozzo for all the technical training and guidance related to imaging. The study was supported by the Swiss National Science Foundation (SNSF) (PCEGP3_186969) and the Canton of Vaud.

## Author contributions

**Lucas Schoenfeldt**: Conceptualization; Data curation; Software; Formal analysis; Visualization; Methodology; Writing—original draft; Project administration; Writing—review and editing. **Patrick T Paine**: Conceptualization; Supervision; Visualization; Writing—original draft. **Sara Picó**: Software; Formal analysis; Visualization; Writing—original draft. **Nibrasul H Kamaludeen M**: Formal analysis; Investigation. **Grace B Phelps**: Formal analysis; Investigation. **Calida Mrabti**: Investigation. **Gabriela Desdín-Micó**: Formal analysis. **María del Carmen Maza**: Formal analysis; Investigation. **Kevin Perez**: Conceptualization; Software; Formal analysis; Supervision; Investigation; Visualization; Methodology; Writing—original draft; Project administration; Writing—review and editing. **Alejandro Ocampo**: Conceptualization; Data curation; Supervision; Funding acquisition; Methodology; Project administration; Writing—review and editing.

Source data underlying figure panels in this paper may have individual authorship assigned. Where available, figure panel/source data authorship is listed in the following database record: biostudies:S-SCDT-10_1038-S44321-025-00265-9.

## Disclosure and competing interests statement

KP and AO are co-founders and shareholders of EPITERNA SA (non-financial interests). AO is the co-founder of Longevity Consultancy Group (non-financial interests).

# Expanded View Figures

**Figure EV1. Optimized cocktail (2c) multiparameter rejuvenation of aging hallmarks are recapitulated in human keratinocytes.**

(**A**, **B**) MTS quantification of cell density following treatment of human epidermal keratinocyte with TCP (**A**) and Repsox (**B**). Red arrows indicate selected concentrations. (**C**) Immunofluorescence and quantification of γH2AX following 2c treatment in keratinocytes (6 days, "6D"). (**D**, **E**) Immunofluorescence and quantification of H3K9me3 (**D**) and H3K27me3 (**E**) following 2c treatment (6D) in keratinocytes. (**F**) SA-beta-gal staining and quantification of senescence following 2c treatment (6D) in keratinocytes. (**G**, **H**) Immunofluorescence and quantification of γH2AX (**G**) senescence-associated beta-galactosidase (SA-beta-gal) (**H**) following doxorubicin (100 nM) treatment in 2c pretreated keratinocytes (6D). Data were mean ± SEM (**A**, **B**, **F**, **H**), median ± IQR (**C**, **E**, **G**). (**A**–**H**) $n \geq 3$. Statistical significance was assessed by comparison to untreated control using paired two-tailed $t$-test (**C**–**F**), one-way ANOVA and Dunnett correction (**G**, **H**).

▶

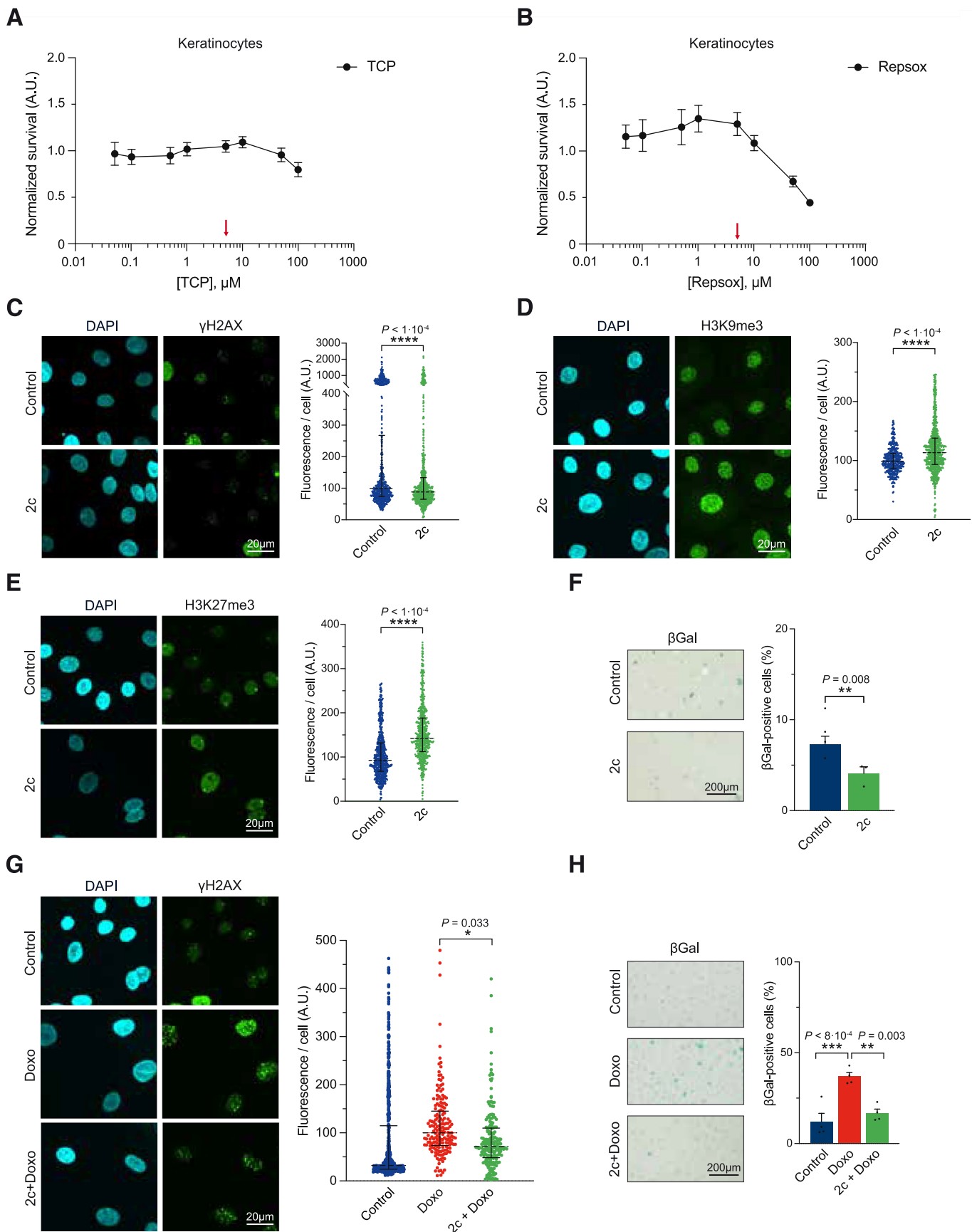

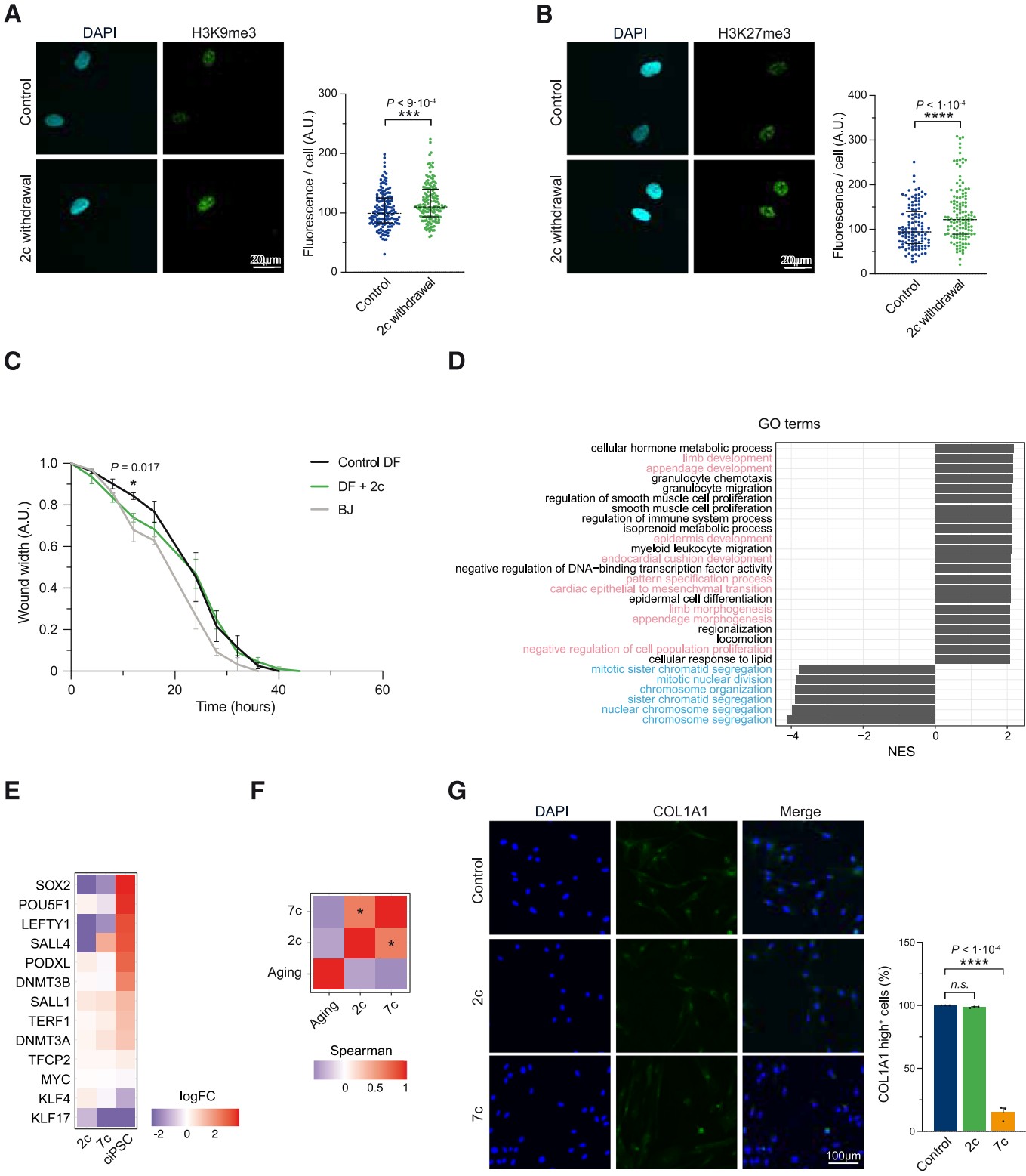

◀

**Figure EV2. Reduced 2c cocktail shows long-lasting effects upon treatment and no dedifferentiation.**

(A, B) Immunofluorescence and quantification of H3K9me3 (A) and H3K27me3 (B) upon 2c treatment for 6 days (6 days, "6D") followed by 30 days chemical withdrawal. (C) Scratch assay analysis upon 2c treatment in adult dermal fibroblasts (DF, 6D) alongside a neonatal control (BJ). (D) Gene ontology (GO) terms overlap between 2c and 7c treatments, with developmental (in pink) and cell cycle (in blue) pathways highlighted. (E) Heatmap showing the expression pattern of differentially expressed genes associated with pluripotency following 2c and 7c treatment, and induction of chemical iPSC (ciPSC). (F) Heatmap of Spearman's correlation of gene expression associated with aging, 2c and 7c. (G) Fluorescence detection and quantification of fibroblast identity marker COL1A1 following 2c and 7c treatments. Data were median ± IQR (A, B), mean ± SEM (C, E). (A–G) $n \geq 3$. Statistical significance was assessed by comparison to untreated control using paired two-tailed $t$-test (A, B), two-way ANOVA mixed-effects analysis and Geisser-Greenhouse correction (C), Spearman's correlation test (F), one-way ANOVA and Dunnett correction (G). NES normalized enrichment scores, FC fold change.

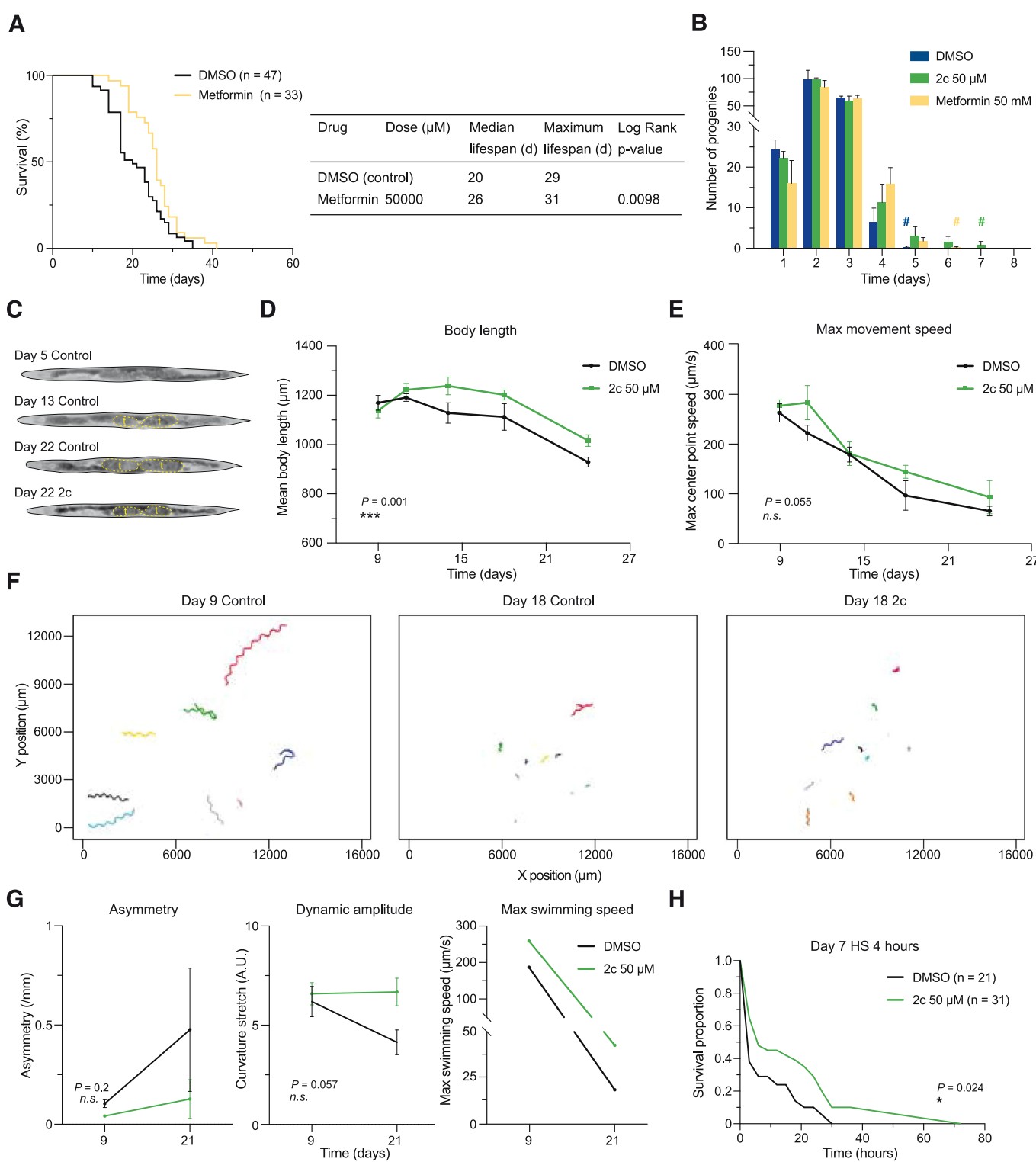

**Figure EV3. Treatment with 2c improves multiple healthspan parameters in *C. elegans*.**

(A) Survival of N2 *C. elegans* upon treatment with Metformin (50 mM). (B) Progeny production and reproductive span (denoted by #) of unmated hermaphrodite *C. elegans* upon treatment with 2c and Metformin. (C) Germline tumoral mass (denoted by t) development at young (day 5), middle (day 13), and old age (day 22) upon 2c treatment in linearized nematodes. (D) Mean body length of N2 *C. elegans* upon treatment with 2c at 50 μM. (E) Maximum movement speed of N2 *C. elegans* upon treatment with 2c at 50 μM. (F) 30-second movement map of N2 *C. elegans* at day 9 and day 18 upon treatment with 2c. (G) Mean swimming asymmetry, amplitude, and max speed of N2 *C. elegans* upon treatment with 2c at 50 μM. (H) Thermotolerance of N2 *C. elegans* to 4 h HS at 7 days of adulthood upon 2c treatment. Data were mean ± SEM (B, D, E, G). (B, D, E) $n \geq 10$, (G) $n \geq 4$. Number of animals and measures are detailed in Methods. Statistical significance was assessed by comparison to vehicle control using log-rank (Mantel-Cox) test (A), by comparison to vehicle control using two-way ANOVA mixed-effects analysis and Geisser-Greenhouse correction (D, E), paired two-tailed *t*-test for day 21 of adulthood (G), log-rank (Mantel-Cox) test (H). HS heat shock.

