## [Peer Review File · EMBO Molecular Medicine]

Chemical reprogramming ameliorates cellular hallmarks of aging and extends lifespan

Alejandro Ocampo, Lucas Schoenfeldt, Patrick Paine, Sara Pico, Nibrasul Kamaludeen Mariyath, Grace Phelps, Calida Mrabti, Gabriela Desdín-Micó, María Maza, and Kevin Perez

Corresponding author(s): Alejandro Ocampo (alejandro.ocampo@unil.ch), Kevin Perez (kevin@epiterna.com)

Review Timeline:

Transfer Date:	25th Mar 25
Editorial Decision:	26th Mar 25
Revision Received:	1st Apr 25
Editorial Decision:	24th Apr 25
Revision Received:	10th Jun 25
Accepted:	11th Jun 25

Editor: Zeljko Durdevic

Transaction Report: This manuscript was transferred to The EMBO Molecular Medicine following peer review at another journal.

26th Mar 2025

Dear Dr. Ocampo,

Thank you for the submission of your manuscript to EMBO Molecular Medicine. Please complete the revision of your manuscript and upload the point-by-point response to the referee concerns.

I look forward to seeing a revised form of your manuscript as soon as possible. Use this link to login to the manuscript system and submit your revision: *Link Unavailable*

Yours sincerely,

Zeljko Durdevic

We require:

- 1) A .docx formatted version of the manuscript text (including legends for main figures, EV figures and tables). Please make sure that the changes are highlighted to be clearly visible.
- 2) Individual production quality figure files as .eps, .tif, .jpg (one file per figure). For guidance, download the 'Figure Guide PDF': (<https://www.embopress.org/page/journal/17574684/authorguide#figureformat>).
- 3) A .docx formatted letter INCLUDING the reviewers' reports and your detailed point-by-point responses to their comments. As part of the EMBO Press transparent editorial process, the point-by-point response is part of the Review Process File (RPF), which will be published alongside your paper.
- 4) A complete author checklist, which you can download from our author guidelines (<https://www.embopress.org/page/journal/17574684/authorguide#submissionofrevisions>). Please insert information in the checklist that is also reflected in the manuscript. The completed author checklist will also be part of the RPF.
- 5) Please note that all corresponding authors are required to supply an ORCID ID for their name upon submission of a revised manuscript.
- 6) It is mandatory to include a 'Data Availability' section after the Materials and Methods. Before submitting your revision, primary datasets produced in this study need to be deposited in an appropriate public database, and the accession numbers and database listed under 'Data Availability'. Please remember to provide a reviewer password if the datasets are not yet public (see <https://www.embopress.org/page/journal/17574684/authorguide#dataavailability>).

- 8) At EMBO Press we ask authors to provide source data for the main manuscript figures. Our source data coordinator will contact you to discuss which figure panels we would need source data for and will also provide you with helpful tips on how to

upload and organize the files.

12) Author contributions: You will be asked to provide CRediT (Contributor Role Taxonomy) terms in the submission system. These replace a narrative author contribution section in the manuscript.

13) A Conflict of Interest statement should be provided in the main text.

14) Every published paper now includes a 'Synopsis' to further enhance discoverability. Synopses are displayed on the journal webpage and are freely accessible to all readers. They include a short stand first (maximum of 300 characters, including space) as well as 2-5 one-sentences bullet points that summarizes the paper. Please write the bullet points to summarize the key NEW findings. They should be designed to be complementary to the abstract - i.e. not repeat the same text. We encourage inclusion of key acronyms and quantitative information (maximum of 30 words / bullet point). Please use the passive voice. Please attach these in a separate file or send them by email, we will incorporate them accordingly.

15) Include a Reagents and Tools Table as part of the Methods section, which can be downloaded from our author guidelines (<https://www.embopress.org/page/journal/17574684/authorguide#structuredmethods>)

1. The authors have trained a transcription clock with only 12 samples. This is not sufficient for model training and additional samples (either generated by the authors or from published datasets) should be added.
2. The authors trained a linear model using only young and old samples. As middle age samples were not included in the training set, a classifier would have been more appropriate.
3. The authors have not stated the error of their transcription clock. This aspect needs to be added to the manuscript.
4. The boxplots for the control sample age predictions in figure 4E are unusually narrow. It is unlikely that a model would perfectly predict all these samples, and this suggests that these samples were included in the training set of this model. The authors need to state whether this is the case or not. Furthermore, if these samples were included in the training set, then the model should be retrained without these samples as otherwise it will be overfitted to the control samples, which makes it very difficult to interpret the results.

Following the comments from Referee #1 (Major 1-4), we have decided to remove the transcriptomic clock results from the manuscript.

As a reminder, as suggested initially by the referee, we first explored Yang JH et al., 2023 (PMID: 37437248) and tried to apply their aging clocks to our dataset. However, the relevant clock was based on an unpublished model referenced in Kriukov D. et al., 2022 (doi: 10.1101/2022.12.12.520058), explicitly stating: “*To estimate the systemic rejuvenation occurring during reprogramming, we utilized our recently developed mouse and human multi-tissue gene expression aging clocks (unpublished).*” Since this clock has not yet been made publicly available, we were unable to use it to assess rejuvenation with 2c/7c. Moreover, unlike DNA methylation clocks, RNA-seq-based transcriptomic clocks are highly susceptible to batch effects, making cross-dataset comparisons challenging.

Given these limitations, we developed our own transcriptomic clock. However, as correctly pointed out by Referee #1, our model had several constraints:

- A small sample size (N = 12), limiting statistical power.
- Uneven age distribution, which impacts model generalizability.
- The inability to separate training and test sets, increasing the risk of overfitting.

Given these challenges, we have now fully removed the transcriptomic clock results from our manuscript. We hope this decision strengthens the rigor of our study and allows for a clearer interpretation of the remaining data.

24th Apr 2025

Dear Dr. Perez,

Thank you for the submission of your manuscript to EMBO Molecular Medicine. We have now received feedback from a reviewer who agreed to re-evaluate your manuscript. I am pleased to inform you that we will be able to accept your manuscript pending the following final amendments:

- 1) Please implement referee's suggestions.
- 2) Authors: E-mail correspondence to Sara Pico and Grace Phelps could not be delivered. Please update their e-mail addresses and make sure to enter correct e-mail addresses for all authors in our submission system.
- 3) Figures: Please upload main figures and EV figures as individual, high-resolution files in TIFF, EPS or PDF format. The legends should be compiled at the end of the manuscript text, with the EV figure legends after the main figure legends and with the heading "Expanded View Figure Legends" Please check "Author Guidelines" for more information:
<https://www.embopress.org/page/journal/17574684/authorguide#figureformat>
<https://www.embopress.org/page/journal/17574684/authorguide#expandedview>
- 4) Tables: Please place the 2 tables in the main manuscript file between the main and EV figure legends.
- 5) Author checklist: Please submit a complete checklist. <https://www.embopress.org/pb-assets/embosite/EMBO%20Press%20Author%20Checklist-1642513524327.xlsx>
- 6) In the main manuscript file, please do the following:
 - Please address all comments suggested by our data editors listed below:
 - o Data availability statement:
 1. Please note that the data availability statement is not provided in the manuscript.
 - o Figure legends:
 1. Please note that the box plots need to be defined in terms of minima, maxima, centre, bounds of box and whiskers, and percentile in the legend of figure 4E.
 2. Please note that information related to n is missing in the legends of figures 5E, F, G, H, I, J; EV3 B, D, E, G; S2A, B.
 3. Please note that n=2 in figures 1B, C; 2A-C; 3D, H; S3A, B, C.
 4. Please note that the error bars are not defined in the legend of figure S1 A.
 5. Please note that the exact p values are not provided in the legends of figures 1B, C, D, E, F; 3A-C, G; EV1 C, D, E, H; EV2 A, B, G.
 6. Please indicate the statistical test used for data analysis in the legends of figures 3A-C.
 - Callouts for "Supplementary Figure 1A" and "Supplementary Figure 1B" should be corrected to the appropriate EV Figure or Appendix Figure.
 - In Methods, provide the statement that informed consent was obtained from all human subjects and confirm that the experiments conformed to the principles set out in the WMA Declaration of Helsinki and the Department of Health and Human Services Belmont Report.
 - In Methods, provide the antibody dilutions that were used for each antibody.
 - Indicate in legends exact n and exact p values, not a range, along with the statistical test used. To keep the figures "clear" some authors found providing an Appendix table Sx with all exact p-values preferable. You are welcome to do this if you want to.
 - Please include structured Methods section that includes a Reagents and Tools Table (should be uploaded as a separate file) followed by a Methods and Protocols section. More information on how to adhere to this format as well as downloadable templates (.docx) for the Reagents and Tools Table can be found in our author guidelines:
<https://www.embopress.org/page/journal/17574684/authorguide#structuredmethods>
An example of a paper with Structured Methods can be found here:
<https://www.embopress.org/doi/full/10.1038/s44320-024-00037-6#sec-4>
 - Rename "Competing interests" to "Disclosure Statement & Competing Interests" and place it after the "Acknowledgements". We updated our journal's competing interests policy in January 2022 and request authors to consider both actual and perceived competing interests. Please review the policy <https://www.embopress.org/competing-interests> and update your competing interests if necessary.
 - o Author contributions: Please remove it from the manuscript and specify author contributions in our submission system. CRediT has replaced the traditional author contributions section because it offers a systematic machine-readable author contributions format that allows for more effective research assessment. You are encouraged to use the free text boxes beneath each contributing author's name to add specific details on the author's contribution. More information is available in our guide to authors:
<https://www.embopress.org/page/journal/17574684/authorguide#authorshipguidelines>
 - Data availability statement should contain information about data that cannot be published in the manuscript itself (e.g. structural data, high-throughput sequencing or data from large-scale gene expression experiments). Raw data from large-scale datasets should be deposited in one of the relevant databases and made freely available prior the publication of the manuscript. Use the following format to report the accession number of your data:

[data type]: [full name of the resource] [accession number/identifier] ([doi or URL or identifiers.org/DATABASE:ACCESSION])

Please check "Author Guidelines" for more information.

<https://www.embopress.org/page/journal/17574684/authorguide#availabilityofpublishedmaterial>

7) Funding: Please merge it with "Acknowledgements".

8) Appendix: Please add title page with table of content and page numbers.

9) The Paper Explained: Please provide "The Paper Explained" and add it to the main manuscript text. Please check "Author Guidelines" for more information. <https://www.embopress.org/page/journal/17574684/authorguide#researcharticleguide>

10) Synopsis: Every published paper now includes a 'Synopsis' to further enhance discoverability. Synopses are displayed on the journal webpage and are freely accessible to all readers. They include separate synopsis image and synopsis text.

- Synopsis image: Please provide a visual abstract as a high-resolution jpeg file 550 px-wide x 200-600 pixels high to illustrate your article.

- Synopsis text: Please provide a short standfirst (maximum of 300 characters, including space) as well as 2-5 one sentence bullet points that summarise the paper as a .doc file. Please write the bullet points to summarise the key NEW findings. They should be designed to be complementary to the abstract - i.e. not repeat the same text. We encourage inclusion of key acronyms and quantitative information (maximum of 30 words / bullet point). Please use the passive voice.

11) As part of the EMBO Publications transparent editorial process initiative (see our Editorial at

<http://embomolmed.embopress.org/content/2/9/329>), EMBO Molecular Medicine will publish online a Review Process File (RPF) to accompany accepted manuscripts. This file will be published in conjunction with your paper and will include the anonymous referee reports, your point-by-point response and all pertinent correspondence relating to the manuscript. Let us know whether you agree with the publication of the RPF and as here, if you want to remove or not any figures from it prior to publication. Please note that the Authors checklist will be published at the end of the RPF.

12) Please provide a point-by-point letter INCLUDING my comments as well as the reviewer's reports and your detailed responses (as Word file).

I look forward to reading a new revised version of your manuscript as soon as possible.

Yours sincerely,

Zeljko Durdevic

*** Instructions to submit your revised manuscript ***

1) a .docx formatted version of the manuscript text (including Figure legends and tables)

2) Separate figure files*

3) supplemental information as Expanded View and/or Appendix. Please carefully check the authors guidelines for formatting

Expanded view and Appendix figures and tables at
<https://www.embopress.org/page/journal/17574684/authorguide#expandedview>

4) a letter INCLUDING the reviewer's reports and your detailed responses to their comments (as Word file).

5) The paper explained: EMBO Molecular Medicine articles are accompanied by a summary of the articles to emphasize the major findings in the paper and their medical implications for the non-specialist reader. Please provide a draft summary of your article highlighting

6) Author contributions: the contribution of every author must be detailed in a separate section.

7) EMBO Molecular Medicine now requires a complete author checklist (<https://www.embopress.org/page/journal/17574684/authorguide>) to be submitted with all revised manuscripts. Please use the checklist as guideline for the sort of information we need WITHIN the manuscript. The checklist should only be filled with page numbers where the information can be found. This is particularly important for animal reporting, antibody dilutions (missing) and exact values and n that should be indicated instead of a range.

8) Every published paper now includes a 'Synopsis' to further enhance discoverability. Synopses are displayed on the journal webpage and are freely accessible to all readers. They include a short stand first (maximum of 300 characters, including space) as well as 2-5 one sentence bullet points that summarise the paper. Please write the bullet points to summarise the key NEW findings. They should be designed to be complementary to the abstract - i.e. not repeat the same text. We encourage inclusion of key acronyms and quantitative information (maximum of 30 words / bullet point). Please use the passive voice. Please attach these in a separate file or send them by email, we will incorporate them accordingly.

You are also welcome to suggest a striking image or visual abstract to illustrate your article. If you do please provide a jpeg file 550 px-wide x 300-600px high.

9) A Conflict of Interest statement should be provided in the main text

10) Please note that we now mandate that all corresponding authors list an ORCID digital identifier. This takes <90 seconds to complete. We encourage all authors to supply an ORCID identifier, which will be linked to their name for unambiguous name identification.

Currently, our records indicate that the ORCID for your account is 0000-0002-1802-185X.

Link Not Available

11) Include a Reagents and Tools Table as part of the Methods section, which can be downloaded from our author guidelines (<https://www.embopress.org/page/journal/17574684/authorguide#structuredmethods>)

Photos 400-800 DPI

*Additional important information regarding figures and illustrations can be found at
<https://bit.ly/EMBOPressFigurePreparationGuideline>. See also figure legend preparation guidelines:
<https://www.embopress.org/page/journal/17574684/authorguide#figureformat>

***** Reviewer's comments *****

Referee #1 (Remarks for Author):

Shoenfeldt et al have appropriately addressed my major concerns from my previous review by removing their transcription clock results. At this time, transcription clocks still face several challenges and it is difficult to apply existing models to new datasets. As these concerns have been addressed, this manuscript should be considered for publication. I just have one minor comment, which wasn't fully addressed on lines 309-310. The statement regarding figure EV2F is a little confusing and potentially could be reworded to "Finally, gene signatures in 2c and 7c were inversely correlated to aging". It would also be helpful to explain in the figure legend which ages were compared against each other to generate the aging-associated gene signature in the figure.

Editor's comments

2) Authors: E-mail correspondence to Sara Pico and Grace Phelps could not be delivered. Please update their e-mail addresses and make sure to enter correct e-mail addresses for all authors in our submission system.

3) Figures: Please upload main figures and EV figures as individual, high-resolution files in TIFF, EPS or PDF format. The legends should be compiled at the end of the manuscript text, with the EV figure legends after the main figure legends and with the heading "Expanded View Figure Legends" Please check "Author Guidelines" for more information:

<https://www.embopress.org/page/journal/17574684/authorguide#figureformat>

<https://www.embopress.org/page/journal/17574684/authorguide#expandedview>

The figure legends and expanded view figure legends have been compiled at the end of the manuscript text with the appropriate headings.

4) Tables: Please place the 2 tables in the main manuscript file between the main and EV figure legends.

The tables have been placed in the main manuscript file between the main and EV figure legends.

5) Author checklist: Please submit a complete checklist. <https://www.embopress.org/pb-assets/embo-site/EMBO%20Press%20Author%20Checklist-1642513524327.xlsx>

We have submitted the Author checklist.

6) In the main manuscript file, please do the following:

- Please address all comments suggested by our data editors listed below:

o Data availability statement:

1. Please note that the data availability statement is not provided in the manuscript.

We have provided a data availability statement.

o Figure legends:

1. Please note that the box plots need to be defined in terms of minima, maxima, centre, bounds of box and whiskers, and percentile in the legend of figure 4E.

2. Please note that information related to n is missing in the legends of figures 5E, F, G, H, I, J; EV3 B, D, E, G; S2A, B.

3. Please note that n=2 in figures 1B, C; 2A-C; 3D, H; S3A, B, C.

4. Please note that the error bars are not defined in the legend of figure S1 A.

5. Please note that the exact p values are not provided in the legends of figures 1B, C, D, E, F; 3A-C, G; EV1 C, D, E, H; EV2 A, B, G.

We added the following information in the legends of Figure 5: "(E-G,J) n \geq 10, (H-I) n \geq 4.", EV3: "(B,D,E) n \geq 10, (G) n \geq 4.", and S2: "(A,B) n=3.". In addition, we described the error bars of figure S1 A: "Data are mean \pm SEM (A)". Finally, we added the following information in the legends regarding the box plots au figure 4E: "Data are median \pm IQR with whiskers extending to the furthest values within 1.5·IQR". Finally, we added the exact p-values in Appendix Table S3 with all exact p-values for Figures 1B, C, D, E, F; 3A-C, G; EV1 C, D, E, H; EV2 A, B, G.

6. Please indicate the statistical test used for data analysis in the legends of figures 3A-C.

We added the following information in the legends of Figure 3: Statistical significance was assessed by comparison to untreated control using paired two-tailed t-test (A-C,F,H-I)”.

- Callouts for "Supplementary Figure 1A" and "Supplementary Figure 1B" should be corrected to the appropriate EV Figure or Appendix Figure.

On lines 154-157, the correct callout was edited: “On the other hand, we noted that the senescence-associated secretory cytokine *IL6* was significantly upregulated upon long-term treatment with 7c and therefore could not conclude a positive effect of 7c treatment on senescence (Fig. 1F and Appendix Fig. S1A).”

On lines 166-170, the correct callout was edited: “Interestingly, numerous cellular reprogramming, stem cell, and self-renewal genes within the GO term developmental pathways were significantly upregulated following 7c treatment compared to control including WNT5A, NOTCH1A, SOX4, SALL1, NOG, and BMP4 indicating that 7c induced a shift towards a developmental associated transcriptomic profile (Fig. 1I and Appendix Fig. S1B).”

- In Methods, provide the statement that informed consent was obtained from all human subjects and confirm that the experiments conformed to the principles set out in the WMA Declaration of Helsinki and the Department of Health and Human Services Belmont Report.

The following addition was made in the Ethics and Consent section of the Methods: “Biological materials used in this study were obtained from Biopredic (France) with appropriate ethical approval. Informed consent was obtained from all donors prior to sample collection. All procedures were conducted in accordance with the ethical standards of the institutional and/or national research committee, the 1964 Declaration of Helsinki and its later amendments, and the principles outlined in the U.S. Department of Health and Human Services Belmont Report.”.

- In Methods, provide the antibody dilutions that were used for each antibody.

The following section was modified to include dilutions that were used for each antibody: “Antibodies were provided from the following companies. Abcam: anti-H3K27me3 (ab192985, 1:400); Cell Signaling: anti-H3K9me3 (13969, 1:400), anti- γ H2AX (9718, 1:400), anti-Ki67 (15580, 1:400), anti-COL1A1 (260043, 1:800); Thermofisher: anti-Rabbit (A32790, 1:800); Roth: DAPI (6843.1, 500ng/mL)”.

- Indicate in legends exact n and exact p values, not a range, along with the statistical test used. To keep the figures "clear" some authors found providing an Appendix table Sx with all exact p-values preferable. You are welcome to do this if you want to.

An appendix table was added containing all exact p-values: “Appendix Table S3. Statistical analyses exact p-values.”

Figure panel	Pair	p-value
Fig. 1B	Control-7c	2.5·10 ⁻¹⁰
Fig. 1C	Control-Doxo	1.3·10 ⁻¹⁰
Fig. 1C	Doxo-Doxo+7c	1.3·10 ⁻¹⁰
Fig. 1D	Control-7c	6.1·10 ⁻³⁸
Fig. 1E	Control-7c	8.7·10 ⁻³²
Fig. 1F	Control-7c (Gadd45b)	1.6·10 ⁻⁵
Fig. 3A	Control-2c	1.8·10 ⁻⁶
Fig. 3B	Control-2c	1.9·10 ⁻²⁸
Fig. 3C	Control-2c	2.0·10 ⁻²⁹
Fig. 3G	Control-2c 6D (p21)	2.6·10 ⁻⁶
Fig. 3G	Control-2c 29D (p21)	2.7·10 ⁻⁶
Fig. 4D	DEG 2c-DEG 7c (Upreg.)	3.1·10 ⁻⁴
Fig. 4D	DEG 2c-DEG 7c (Downreg.)	7.8·10 ⁻⁸⁴
Fig. 4D	DEG 2c-DEG 7c (All)	1.3·10 ⁻¹⁸
Fig. EV1C	Control-2c	1.1·10 ⁻⁵
Fig. EV1D	Control-2c	3.2·10 ⁻⁸
Fig. EV1E	Control-2c	2.2·10 ⁻⁴⁵
Fig. EV1H	Control-Doxo	8.0·10 ⁻⁴
Fig. EV2A	Control-2c withdrawal	9.0·10 ⁻⁴
Fig. EV2B	Control-2c withdrawal	1.1·10 ⁻²²
Fig. EV2D	Control-7c	4.1·10 ⁻⁷

Appendix Table S3. Statistical analyses exact p-values.

- Please include structured Methods section that includes a Reagents and Tools Table (should be uploaded as a separate file) followed by a Methods and Protocols section. More information on how to adhere to this format as well as downloadable templates (.docx) for the Reagents and Tools Table can be found in our author

guidelines: <https://www.embopress.org/page/journal/17574684/authorguide#structuredmethods>

An example of a paper with Structured Methods can be found here:

<https://www.embopress.org/doi/full/10.1038/s44320-024-00037-6#sec-4>

We have included a Reagents and Tools Table attached as a separate file.

- Rename "Competing interests" to "Disclosure Statement & Competing Interests" and place it after the "Acknowledgements". We updated our journal's competing interests policy in January 2022 and request authors to consider both actual and perceived competing interests. Please review the policy <https://www.embopress.org/competing-interests> and update your competing interests if necessary.

The "Competing interests" section has been renamed and placed accordingly.

- Author contributions: Please remove it from the manuscript and specify author contributions in our submission system. CRediT has replaced the traditional author contributions section because it offers a systematic machine-readable author contributions format that allows for more effective research assessment. You are encouraged to use the free text boxes beneath each contributing author's name to add specific details on the author's contribution. More information is available in our guide to authors:

<https://www.embopress.org/page/journal/17574684/authorguide#authorshipguidelines>

We have updated the author contribution directly on the website as requested.

- Data availability statement should contain information about data that cannot be published in the manuscript itself (e.g. structural data, high-throughput sequencing or data from large-scale gene expression experiments). Raw data from large-scale datasets should be deposited in one of the relevant databases and made freely available prior the publication of the manuscript. Use the following format to report the accession number of your data:

The datasets produced in this study are available in the following databases:
[data type]: [full name of the resource] [accession number/identifier] ([doi or URL
or identifiers.org/DATABASE:ACCESSION])

Please check "Author Guidelines" for more information. <https://www.embopress.org/page/journal/17574684/authorguide#availabilityofpublishedmaterial>

We have provided a Data availability statement in the Methods section: "The raw data (FASTQ) and count data (logCPM) for the RNA-seq profiling of human dermal fibroblasts treated with 2c and 7c have been deposited to Gene Expression Omnibus (GEO), accession number GSE297984."

7) Funding: Please merge it with "Acknowledgements".

The following section was merged: "Acknowledgements and funding. We would like to thank all members of the Ocampo laboratory for helpful discussions. In addition, we would like to thank the UNIL Cellular Imaging Facility and especially Luigi Bozzo for all technical training and guidance related to imaging. The study was supported by the Swiss National Science Foundation (SNSF) and the Canton Vaud."

8) Appendix: Please add title page with table of content and page numbers.

The title page with table of content and page numbers was added to the Appendix.

9) The Paper Explained: Please provide "The Paper Explained" and add it to the main manuscript text. Please check "Author Guidelines" for more information. <https://www.embopress.org/page/journal/17574684/authorguide#researcharticleguide>

The following section has been added to the paper:
"The Paper Explained."

Problem

As aging and age-related diseases place an increasing burden on society, there is an urgent need to develop effective and innovative strategies to mitigate their impact. Cellular reprogramming has emerged as a promising approach to target aging at its root, capable of reversing molecular and functional hallmarks of aging while extending both lifespan and healthspan. However, current reprogramming methods rely on genetic manipulation, limiting the translational potential and raising significant safety concerns including the risk of tumorigenesis and loss of cellular function.

Results

Here, we demonstrate for the first time that partial chemical reprogramming using a seven-compound cocktail (7c) can induce multiparametric rejuvenation across key hallmarks of aging. We further identified an optimized two-compound cocktail (2c) that is sufficient to improve additional aging phenotypes in vitro. Finally, application of this reduced reprogramming cocktail significantly improved multiple markers of

aging, stress-resistance, and healthspan in *C. elegans*, leading to a median lifespan extension of over 42% *in vivo*.

Impact

This work establishes a novel chemical approach to partial cellular reprogramming, offering a clinically viable alternative to genetic methods. By demonstrating conserved rejuvenation effects in both human cells and a whole-organism model, our findings provide a foundation for the development of a novel pharmacological intervention that targets aging at its root.”

10) Synopsis: Every published paper now includes a 'Synopsis' to further enhance discoverability. Synopses are displayed on the journal webpage and are freely accessible to all readers. They include separate synopsis image and synopsis text.

- Synopsis image: Please provide a visual abstract as a high-resolution jpeg file 550 px-wide x 200-600 pixels high to illustrate your article.

- Synopsis text: Please provide a short standfirst (maximum of 300 characters, including space) as well as 2-5 one sentence bullet points that summarise the paper as a .doc file. Please write the bullet points to summarise the key NEW findings. They should be designed to be complementary to the abstract - i.e. not repeat the same text. We encourage inclusion of key acronyms and quantitative information (maximum of 30 words / bullet point). Please use the passive voice.

The synopsis has been added as a separate .doc file submitted with the paper and shows as follows:

“This study reveals that partial chemical reprogramming with defined small-molecule cocktails rejuvenates aged human cells and significantly extends lifespan and healthspan in *C. elegans*, offering a non-genetic strategy to reverse aging phenotypes.

- A seven-compound (7c) reprogramming cocktail was shown to reverse multiple aging hallmarks in human dermal fibroblasts.

- A reduced two-compound (2c) cocktail was identified that retained rejuvenating effects *in vitro* and was sufficient to ameliorate additional aging phenotypes, including senescence, heterochromatin loss, genomic instability, and oxidative stress.

- 2c treatment in *C. elegans* improved stress resistance, thermotolerance, reproductive and healthspan markers, and extended median lifespan by over 42%.

- The results suggest that partial chemical reprogramming could modulate the underlying mechanisms of aging and reproductive aging, offering a potential strategy for extending both healthspan and overall lifespan and healthspan in aging populations.”

The figure synopsis has been added as a separate pdf file.

11) As part of the EMBO Publications transparent editorial process initiative (see our Editorial at <http://embomolmed.embopress.org/content/2/9/329>), EMBO Molecular Medicine will publish online a Review Process File (RPF) to accompany accepted manuscripts. This file will be published in conjunction with your paper and will include the anonymous referee reports, your point-by-point response and all pertinent correspondence relating to the manuscript. Let us know whether you agree with the publication of the RPF and as here, if you want to remove or not any figures from it prior to publication. Please note that the Authors checklist will be published at the end of the RPF.

We agree with the publication of the RPF.

Reviewer's comments

Schoenfeldt et al have appropriately addressed my major concerns from my previous review by removing their transcription clock results. At this time, transcription clocks still face several challenges and it is difficult to apply existing models to new datasets. As these concerns have been addressed, this manuscript should be considered for publication. I just have one minor comment, which wasn't fully addressed on lines 309-310. The statement regarding figure EV2F is a little confusing and potentially could be reworded to "Finally, gene signatures in 2c and 7c were inversely correlated to aging". It would also be helpful to explain in the figure legend which ages were compared against each other to generate the aging-associated gene signature in the figure.

On lines 300-301, the following correction was made in the text: "Finally, gene signatures in 2c and 7c were inversely correlated to aging (Fig. EV2F).".

11th Jun 2025

Dear Dr. Ocampo,

We are pleased to inform you that your manuscript is accepted for publication and is now being sent to our publisher to be included in the next available issue of EMBO Molecular Medicine.

Zeljko Durdevic
Senior Editor
EMBO Molecular Medicine
